# Increased influence of prior choices on perceptual decisions in autism

**Helen Feigin[1†], Shir Shalom-Sperber[1†], Ditza A Zachor[2,3], Adam Zaidel[1]\***

[1]Gonda Multidisciplinary Brain Research Center, Bar Ilan University, Ramat Gan, Israel; [2]The Autism Center/ALUT, Shamir Medical Center, Zerifin, Israel; [3]Sackler Faculty of Medicine, Tel Aviv University, Tel Aviv, Israel

**Abstract** Autism spectrum disorder (ASD) manifests sensory and perceptual atypicalities. Recent theories suggest that these may reflect a reduced influence of prior information in ASD. Some studies have found reduced adaptation to recent sensory stimuli in ASD. However, the effects of prior stimuli and prior perceptual choices can counteract one-another. Here, we investigated this using two different tasks (in two different cohorts): (i) visual location discrimination and (ii) multisensory (visual-vestibular) heading discrimination. We fit the data using a logistic regression model to dissociate the specific effects of prior stimuli and prior choices. In both tasks, perceptual decisions were biased toward recent choices. Notably, the 'attractive' effect of prior choices was significantly larger in ASD (in both tasks and cohorts), while there was no difference in the influence of prior stimuli. These results challenge theories of reduced priors in ASD, and rather suggest an increased consistency bias for perceptual decisions in ASD.

## Introduction

Autism spectrum disorder (ASD) is primarily characterized by impairments in social skills and by restricted and repetitive patterns of behavior (*American Psychiatric Association, 2013*). Although increased or decreased responsiveness to sensory input is also part of the current diagnostic criteria, the nature of atypical perceptual function in ASD is not fully understood, and not part of the diagnostic criteria. Furthermore, the connection between altered perceptual function and the core behavioral symptoms of ASD is unclear.

Classic theories suggest that atypical perceptual function in ASD reflects a shift in favor of processing local (or low-level) sensory details, over integration of those details into a global (or high-level) percept. This may reflect (non-exclusively): impaired global processing (*Shah and Frith, 1983*; *Frith and Happé, 1994*; *Behrmann et al., 2006*; *Booth and Happé, 2018*), superior local processing (*Plaisted et al., 2003*; *Mottron et al., 2006*; *Mottron and Burack, 2021*), and/or a preference for detail (*Plaisted et al., 1999*; *Happé and Frith, 2006*; *Koldewyn et al., 2013*). In recent years, it has been proposed that the bias toward low-level processing in ASD, as well as other perceptual and sensory atypicalities, such as hyper- or hypo-sensitivity to sensory stimuli (*Rogers and Ozonoff, 2005*), may be explained by altered perceptual inference (*Pellicano and Burr, 2012*; *Friston et al., 2013*; *Lawson et al., 2014*).

Perceptual inference has been broadly studied in typical brain function within the Bayesian framework (*Knill and Richards, 1996*; *Kersten and Yuille, 2003*; *Knill and Pouget, 2004*; *Mamassian et al., 2018*). The Bayesian approach provides a principled (computational) account of perception as a process of statistical inference, and posits that top-down expectations ('priors') are integrated together with bottom-up sensory signals (which are inherently 'noisy', that is, variable) to improve perceptual reliability. Although integrating 'priors' reduces sensory noise in favor of high-level (global) perception, this might be at the expense of low-level sensory accuracy. Accordingly, *Pellicano and Burr, 2012* proposed that individuals with ASD might use priors to a lesser degree,

*For correspondence:
adam.zaidel@biu.ac.il

†These authors contributed equally to this work

Competing interests: The authors declare that no competing interests exist.

resulting in superior local processing and altered sensory responsiveness. However, relatively greater sensory (vs. prior) weighting might actually be expected from Bayesian theory due to superior low-level sensory function in ASD (*Brock, 2012*; *Karvelis et al., 2018*). Thus, whether or not priors are altered and what aspects of perceptual inference may differ in ASD requires investigation.

Priors are complex – they reflect a compound (subjective) aggregate of an individual's beliefs about the probability of observing various events or stimuli in the world (*Ma, 2019*). This is influenced by context, statistics of the environment, and other factors, across broad and narrow time-scales. For example, one's prior for hearing explosions might temporarily increase when watching an action movie (but return to normal when leaving the cinema). By contrast, the prior that objects fall to the ground (due to gravity) might be more long-term and constant. It is unclear therefore what aspects of priors might be affected in ASD. At minimum, these can be divided according to their recency: short-term (recent information such as context or immediately preceding events) and long-term (such as beliefs and expectations about the world that were learned throughout life).

Recent studies have found that long-term priors seem intact in ASD. The 'light-from-above prior' (that light more frequently comes from above) was applied by individuals with ASD (*Croydon et al., 2017*), the influence of object size and brightness on perceptions of heaviness were similar in ASD (*Buckingham et al., 2016*; *Hadad and Schwartz, 2019*), and a slow-speed prior was integrated in ASD (*Noel et al., 2020*) – like controls. These results indicate that sensory predictions based on statistical information gathered over long-term experience are indeed built and integrated in ASD. Also, an experiment that tested 'Mooney' images (difficult to interpret pictures, comprising black and white patches) showed that prior exposure to the high-fidelity images helped individuals with ASD as much as controls (*Van de Cruys et al., 2018*), indicating that priors over a time-course of minutes are also used in ASD. Thus, an overall reduction of all priors in ASD does not seem likely.

By contrast, studies that found reduced adaptation to prior stimuli in ASD seem to provide evidence for a reduced impact of *recent* sensory information (in the range of seconds) on perception in ASD. For example: prolonged exposure to a specific facial identity, which typically biases subsequent perception away from that image, did so to a lesser degree in ASD (*Pellicano et al., 2007*). Also, adaptation to numerosity judgements (*Turi et al., 2015*), to loudness (*Lawson et al., 2015*), and to audio-visual asynchronies (*Noel et al., 2017*) was reduced in ASD. At face value, these results seem to support the notion of reduced use of recent prior information in ASD. However, 'adaptation' (like in these studies) most commonly refers to the phenomenon by which exposure to a stimulus repels subsequent perception away from that stimulus (often called a 'negative' effect) – opposite to a Bayesian prior, which generally attracts perception towards the more frequent stimuli (positive effect; although Bayesian priors can also lead to negative effects, for example *Peters et al., 2016*). Thus, while reduced adaptation in ASD may suggest a smaller impact of prior stimuli, this does not imply reduced priors. Also, adaptation is an umbrella term that includes a repertoire of effects (*Kohn, 2007*) – what aspects thereof are specifically altered in ASD requires investigation.

'Serial dependence', a related topic of high recent interest, also studies the impact of recent sensory information (in the range of seconds) on subsequent perception. Specifically, it describes a positive effect by which perceptual decisions are biased toward recently experienced stimuli (*Fischer and Whitney, 2014*; *Liberman et al., 2014*; *Taubert et al., 2016*; *Alexi et al., 2018*; *Liberman et al., 2018*) – opposite to the negative effect of adaptation. Here, too initial evidence shows reduced serial dependence in ASD (*Molesworth et al., 2015*; *Lieder et al., 2019*), suggesting a smaller impact of recent stimuli in ASD.

However, these studies of reduced serial dependence and reduced adaptation in ASD only take into account the previous stimuli, and not the previous choices. This is important because there is mounting evidence that prior perceptual decisions (rather than sensory stimuli) elicit the positive effect of serial dependence (*Kaneko and Sakai, 2015*; *Talluri et al., 2018*; *Feigin et al., 2021*). Thus, the overall impact of recent sensory experience might reflect a composite effect, for example of repulsion away from the prior stimuli (adaptation) and attraction toward the prior choices (serial dependence) which can, at least partially, cancel one-another (*Fritsche et al., 2017*; *Pascucci et al., 2019*). Therefore, understanding specifically which response (to recent prior stimuli or prior choices) is altered in ASD may change our understanding of these results profoundly.

The effect of prior choices is a factor that has been less considered so far in ASD. A recent study of high-level (executive) decision-making found greater consistency in the decisions of individuals with ASD, which may suggest an ASD tendency toward repetitive behavior also in decision-making

(*Wu et al., 2018*). However, this study used a gambling task, which may be different from perceptual decision-making, and thus did not dissociate the serial effects of prior sensory from prior choice information, which needs to be tested together in a perceptual decision-making task. We hypothesized, based on the studies described above, that a positive effect of serial dependence from prior choices may be enlarged in ASD, and that this may cancel out adaptation effects, which could appear as reduced adaptation in ASD if the effects of prior stimuli and choices are not accounted for separately.

In this study, we aimed to dissociate how recent prior stimuli and choices (over a time-course of seconds) affect perceptual decisions in ASD. For this purpose, we tested the performance of individuals with ASD and controls in a visual location discrimination task designed to induce short-term biases over several trials (*Feigin et al., 2021*). In addition, we reanalyzed data from a previous study of multisensory (visual-vestibular) heading direction discrimination performed on a different group of participants (*Zaidel et al., 2015*). Both tasks comprised a two alternative forced choice (2AFC) regarding a single interval stimulus. For the location discrimination task, participants discriminated whether a visual stimulus (solid circle, presented on a screen) lay to the left or right of the screen center. For heading discrimination, participants discriminated whether a linear self-motion stimulus was leftward or rightward of straight ahead. We fit the data with a logistic regression model that separated between the effects of prior stimuli and prior choices.

Strikingly, in both tasks (two different cohorts), we found a stronger effect of prior choices on perceptual decisions in ASD vs. controls. By contrast, individuals with ASD demonstrated a similar effect of adaptation to previous stimuli compared to controls (in both tasks). These results challenge theories of reduced priors in ASD, and rather implicate an increased influence of prior choices in ASD. This may expose a novel aspect of repetitive behavior in ASD, reflected in perceptual decisions.

## Results

The main aim of this study was to investigate whether the influence of recent prior information on perceptual decision-making is altered in ASD. This was tested on two different datasets: (i) visual location discrimination (collected for this study), and (ii) multisensory visual-vestibular heading discrimination (reanalysis of previously published data, *Zaidel et al., 2015*). In both datasets, we found a stronger effect of recent prior information on perceptual decisions in ASD compared to controls. Moreover, we found that this effect was mediated by a greater influence of prior choices in ASD. Specifically, perceptual decisions were more likely to be the same as previous choices (an increased 'consistency' bias). Interestingly, no differences were found regarding the influence of recent sensory information.

### Location discrimination

In the visual location discrimination task, the participants were required to report whether the visual stimulus (a filled circle) lay to the right or to the left of the screen center (2AFC). Stimulus locations were biased to the right or to the left for several 'prior' discriminations, followed by a single unbiased (i.e. balanced) test stimulus (*Figure 1*; rightward and leftward biased trials were interleaved). We first tested the overall effect of these short-term priors by sorting the test stimuli according to their prior bias (right/left) and plotting separate psychometric functions for each prior type (per participant).

### Larger effect of recent priors in ASD for location discrimination

Example psychometric plots for one participant with ASD and one control participant (red and blue shades, left and right subplots of *Figure 2A*, respectively) reveal that for both participants test stimulus discriminations were influenced by recent priors. This is seen by a shift in the psychometric curves for rightward vs. leftward biased priors (light vs. dark shades, respectively). Note that all psychometric curves are plotted as a function of the test stimulus location (which was unbiased). Thus, any differences between the light- and dark-shaded curves reflect different responses to the same test stimuli (therefore shifts in the psychometric curves expose the effects of recent priors).

To understand the direction of the shift – it is easiest to compare the psychometric curves where they cross x = 0° (vertical dotted line). For both participants, the lighter shaded curve (right biased priors) crosses at a higher value vs. the darker curve (left biased priors). This means that there is a

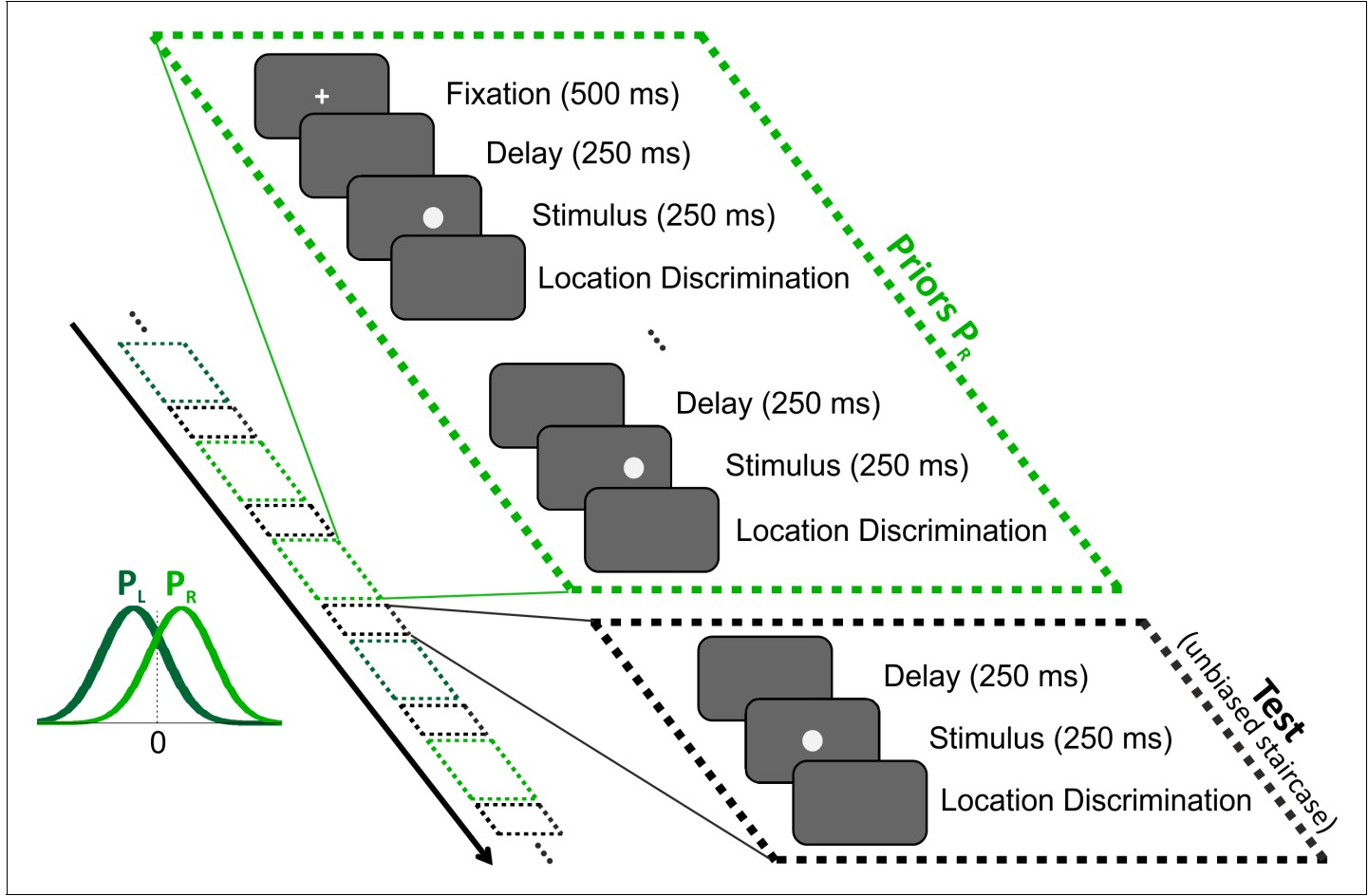

**Figure 1.** Location discrimination – event sequence within and across trials. All stimuli (filled circles) were discriminated according to location, that is, whether the circle lay to the left or to the right of the screen center (2AFC). Each trial comprised a series of stimuli and discriminations that began with a fixation point (at the screen center), followed by a series of 'prior' stimuli biased to the right (light green) or left (dark green), and then by a single (unbiased) 'test' stimulus. The black arrow marks the timecourse across trials, which interleaved trials with right and left prior types (light and dark green dashed boxes, respectively) pseudorandomly, each ending with a test stimulus (black dashed boxes). Prior circle locations were drawn from a normal distribution either biased to the right or left of the screen center (light and dark green probability distributions, $P_R$ and $P_L$, respectively). Test circle locations followed an unbiased staircase procedure.

higher probability for choosing right following rightward biased priors (and a higher probability for choosing left following leftward biased priors). Another way to view and to quantify this is by the difference between the curves' PSEs (where they cross the horizontal dotted line y = 0.5). A larger PSE difference ($\Delta$PSE, see *Equation 4* in the Materials and methods) indicates a stronger effect, and positive $\Delta$PSE values (like these examples) mean that the participant's choices were biased in the direction of the recent priors. In these examples, the $\Delta$PSE was larger for the participant with ASD (difference between the red-shaded curves) versus the control participant (difference between the blue-shaded curves).

At the group level, the $\Delta$PSEs were highly significant for the ASD group (red bar, *Figure 2B*; $t_{17}$ = 5.2, p = 6.7$\cdot$10$^{-5}$, *Cohen's d* = 1.23, 95% CI = [0.6, 1.84], *t*-test), and smaller, but still (albeit marginally) significant for the control group (blue bar, *Figure 2B*; $t_{19}$ = 2.1, p = 0.047 *Cohen's d* = 0.47, 95% CI = [0.006, 0.93], *t*-test). Notably, the $\Delta$PSEs were significantly larger for the ASD vs. control groups ($t_{36}$ = 2.1, p = 0.043, *Cohen's d* = 0.68, 95% CI = [0.02, 1.3], *t*-test). Hence, not only was a significant effect of recent priors seen also in ASD, it was *larger* than that of controls. This result (and the results presented below) does not result from a difference in response times between the groups, because these were similar: mean ± *SD* = 0.73 ± 0.11 s for ASD and 0.78 ± 0.13 s for controls

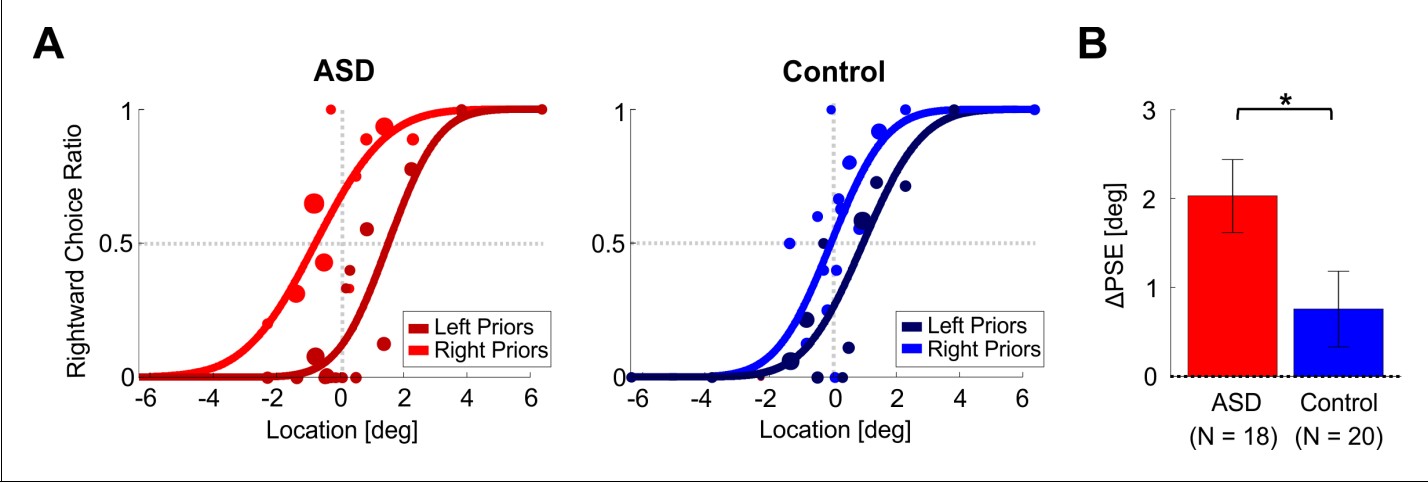

**Figure 2.** Location discrimination – psychometric function shifts. (**A**) Psychometric curves for an example participant with ASD (left plot) and an example control participant (right plot) performing location discrimination. The psychometric curves represent the ratio of rightward choices as a function of stimulus location, sorted according to the prior type (left or right bias, represented by dark and light colors, respectively). The data (circles) were fit with cumulative Gaussian distribution functions (solid lines). Circle size reflects the number of trials for a given stimulus location and prior type. (**B**) PSE shifts (ΔPSE = PSE for left biased priors minus PSE for right biased prior). Red and blue bars represent ΔPSEs (mean ± SEM) for participants with ASD and controls, respectively. *p < 0.05.

The online version of this article includes the following figure supplement(s) for figure 2:

**Figure supplement 1.** Test stimulus convergence for the location discrimination experiment, primary condition.

(calculated across all discriminations, that is, pooling prior and test discriminations; $t_{36} = -1.3$, p = 0.2, *Cohen's d* = 0.42, 95% CI = [−1.06, 0.23], *t*-test).

## Enhanced influence of recent choices in ASD for location discrimination

The above (ΔPSE) analysis suggests that individuals with ASD have a larger influence of recent priors. However, it does not address the underlying reason – specifically, it does not dissociate the separate influences of prior stimuli from prior choices, which may be different. Recent studies have shown that the influence of recent prior stimuli and choices might be in opposite directions – that is, the effects or prior stimuli could be adaptive (reflected by a negative shift), whereas recent choices may lead to a consistency bias (*Kaneko and Sakai, 2015*; *Talluri et al., 2018*; *Pascucci et al., 2019*; *Feigin et al., 2021*). Thus, the effects of prior stimuli and choices could cancel each other out. Furthermore, a larger ΔPSE shift in ASD could result from a weaker stimulus adaptation effect, or a stronger influence of prior choices (or a combination thereof). Hence, conclusions cannot be drawn from the ΔPSE results alone.

To address this, we fit a logistic regression model (per participant) that predicted each participant's discriminations of test stimuli, based on four factors: the current stimulus, previous stimuli, previous choices and a general subjective baseline bias (*Figure 3*). The fourth regression coefficient (baseline bias) was not part of the group comparison because this analysis aimed to investigate the difference in priors, and there was no expected difference in baseline bias (it was, however, still needed for the model fits in order to better estimate the other parameters). We confirmed that the baseline coefficients ($\beta_0$) were indeed not significantly different between the groups ($t_{36} = -1.46$, p = 0.15, *Cohen's d* = −0.47, 95% CI = [−1.1, 0.18], *t*-test). A comparison of the other three regression coefficients revealed a significant difference between the ASD and control groups ($F_{3,34} = 3.3$, p = 0.03, $\eta_p^2 = 0.23$, one-way MANOVA).

Further investigation into this group difference (to assess whether individuals with ASD were affected differently by the previous stimuli or the previous choices), using pairwise comparisons, found that the influence of previous choices ($\beta_{prev\_choices}$ coefficients) was significantly larger in ASD vs. controls (*Figure 4*, center plot; $t_{36} = 2.5$, p = 0.034, *Cohen's d* = 0.8, 97.5% CI = [0.05, 1.57], *t*-test, p-value reported here was multiplied by two to correct for multiple comparisons, see *Statistics* section in the Materials and methods for more details). Thus, while both groups were influenced by

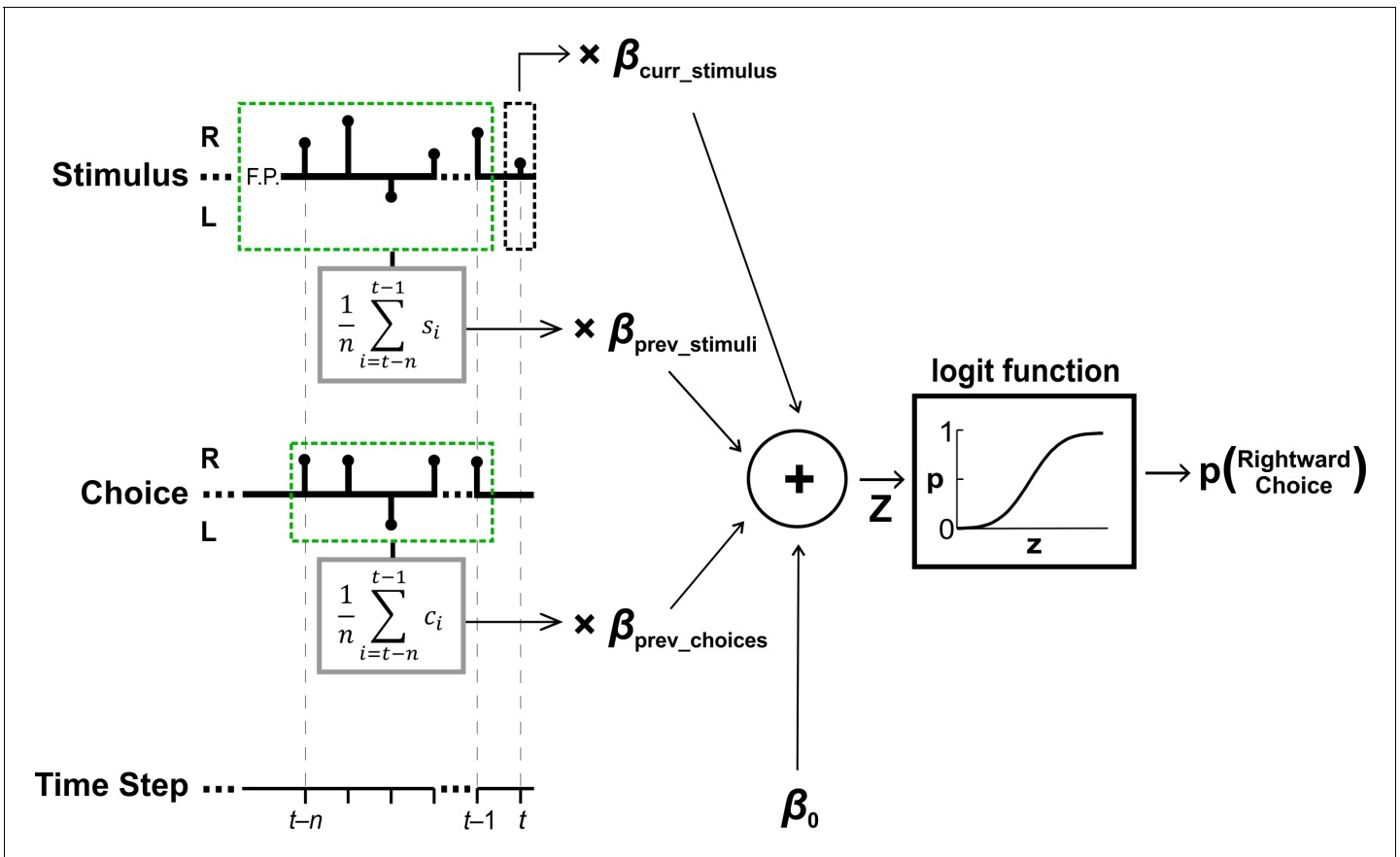

**Figure 3.** Perceptual decision model schematic. Perceptual choices were modeled using a logistic regression with four factors (providing their respective fitted beta coefficients): (i) the current stimulus ($\beta_{curr\_stimulus}$), (ii) previous stimuli ($\beta_{prev\_stimuli}$), (iii) previous choices ($\beta_{prev\_choices}$) and (iv) baseline bias ($\beta_0$). Black stems on the stimulus and choice axes depict the stimuli ($s$) and choices ($c$) for an example trial (indexed by time step), with a test stimulus at time $t$ (black dashed box) and prior stimuli and choices $n$ steps back (light green dashed boxes). The sum of the product of the factors with their respective beta coefficients ($z$) is passed through a logistic function to yield the probability of making a rightward choice in response to the current (test) stimulus at time $t$.

their previous choices in a 'positive' (attractive) manner ($\beta_{prev\_choices}$ coefficients were positive and significant for each group individually: $t_{17} = 9.9$, $p = 2 \cdot 10^{-8}$, Cohen's $d = 2.3$, 95% CI = [1.42, 3.23] and $t_{19} = 6.4$, $p = 4 \cdot 10^{-6}$, Cohen's $d = 1.43$, 95% CI = [0.8, 2] for ASD and controls, respectively; $t$-tests) this effect was significantly increased in ASD. This not only negates hypotheses of reduced priors in ASD – it shows the opposite – an increased influence of prior choices in ASD.

By contrast, the influence of previous stimuli ($\beta_{prev\_stimuli}$ coefficients) was small (**Figure 4**, right plot; to enable comparison between choice and stimulus coefficients, stimulus values were normalized before model fitting to have RMS = 1, like choice). For both groups, the influence of previous stimuli tended to be negative, suggesting an adaptive effect. This was significant for the control group ($t_{19} = -2.5$, $p = 0.021$, Cohen's $d = 0.57$, 95% CI = [−1.03,–0.09], $t$-test) and close to significant for the ASD group ($t_{17} = -1.99$, $p = 0.063$, Cohen's $d = 0.47$, 95% CI = [−0.95, 0.03], $t$-test). There was no significant difference between groups (**Figure 4**; $t_{36} = 0.01$, $p = 0.99$, Cohen's $d = 0.003$, 95% CI = [−0.63, 0.64], $t$-test; the raw p-value is reported here before correction for multiple comparisons).

Lastly, as expected, the current stimulus coefficients ($\beta_{curr\_stimulus}$) were large, positive and significant for both groups (**Figure 4**, left; $t_{17} = 9.2$, $p = 5 \cdot 10^{-8}$, Cohen's $d = 2.18$, 95% CI = [1.3, 3.03] and $t_{19} = 8.1$, $p = 1.5 \cdot 10^{-7}$, Cohen's $d = 1.8$, 95% CI = [1.08, 2.51] for the ASD and control groups, respectively; $t$-tests). Also, these did not differ significantly between groups ($t_{36} = -0.85$, $p = 0.4$, Cohen's $d = 0.28$, 95% CI = [-0.92, 0.37], $t$-test). This indicates that sensitivity did not differ between

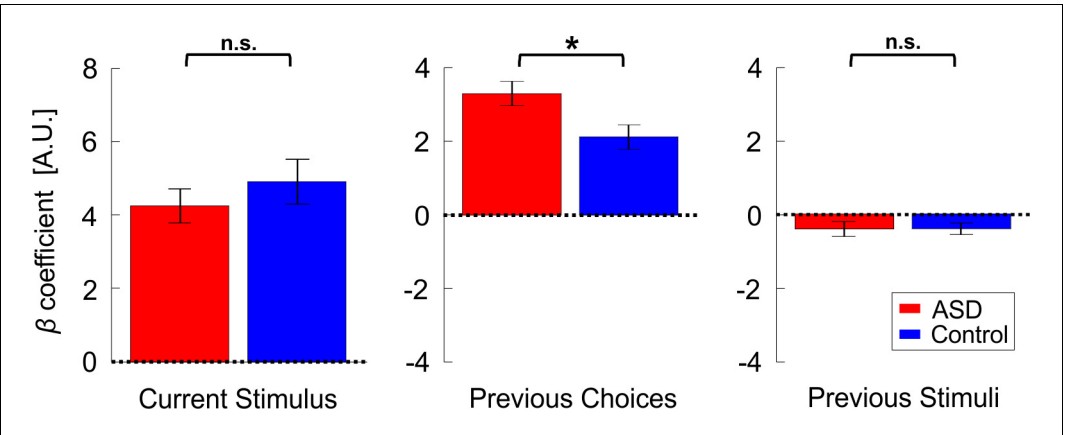

**Figure 4.** Location discrimination (primary condition) – model parameters. Beta coefficients for model parameters: current stimulus, previous choices and previous stimuli (left, center, and right plots, respectively). Red and blue bars represent the values (mean ± SEM) for participants with ASD and controls, respectively. *p <0.05, n.s. – not significant.

The online version of this article includes the following figure supplement(s) for figure 4:

**Figure supplement 1.** Location discrimination (primary condition) – five choices back model parameters.
**Figure supplement 2.** Scatter of $\beta_{prev\_choices}$ coefficients (primary condition) vs. ADOS scores.

the groups. We confirmed this by comparing thresholds from the psychometric functions (see Materials and methods), which also did not differ significantly between groups ($t_{36}$ = 0.96, p = 0.34, *Cohen's d* = 0.31, 95% CI = [-0.33, 0.95], *t*-test; $BF_{01}$ = 2.21). Hence, the only difference we found in ASD was an increased influence of previous choices. Lastly, stimulus sensitivity ($\beta_{curr\_stimulus}$) did not correlate significantly with the prior choice effect $\beta_{prev\_choices}$ ($r_{36}$ = -0.18, p = 0.27).

## Prior choice effect seen two steps back

To assess whether there is a differential contribution of 'priors', we reran the regression analysis, modified to study the separate contributions of the five preceding 'priors'. There were two potential obstacles to performing this analysis: (a) the number of priors for a given trial was random (mean = 5, see Materials and methods) so there could be fewer than five (there could even be only one prior). (b) Modeling both prior choices and prior stimuli five steps back means 10 'prior' parameters. This could potentially lead to overfitting and spurious results (especially for the more distant ones, where the prior effect is presumably weaker). To overcome these issues, we took the following two measures: (i) we fit the model to all the data (prior and test, viewed as one long sequence of trials) using the five preceding discriminations as 'priors' (irrespective of whether they were initially termed 'prior' or 'test'). This was possible only in the primary location discrimination condition (presented thus far) because the stimulus color and task (i.e. which buttons to use) were the same for both prior and test circles (it was not possible in the *Response Invariant* condition, presented below, in which these parameters differed). (ii) We first looked whether the effect of prior choices and prior stimuli were significant for the most recent two discriminations (*t*-1 and *t*-2): both prior choices were significant (p < 6·10⁻⁵; *t*-tests) and both prior stimuli were not (p > 0.14; *t*-tests). Thus, to prevent overfitting/ spurious results, we investigated further only the prior choice effect for the five preceding discriminations (*Figure 4—figure supplement 1*).

We found that two choices back (*t*-1 and *t*-2) significantly (positively) affected decisions (*Figure 4—figure supplement 1B*; for one step back (*t*-1): $t_{17}$ = 7.16, p = 1.6·10⁻⁶, *Cohen's d* = 1.69, 95% CI = [0.95, 2.41] for the ASD group and $t_{19}$ = 4.46, p = 0.0003, *Cohen's d* = 1.00, 95% CI = [0.45, 1.53] for the control group. For two steps back (*t*-2): $t_{17}$ = 9.09, p = 6·10⁻⁸, *Cohen's d* = 2.14, 95% CI = [1.28, 2.98] for the ASD group and $t_{19}$ = 4.63, p = 0.0002, *Cohen's d* = 1.03, 95% CI = [0.48, 1.57] for the control group, *t*-tests). Beyond that (*t*-3, *t*-4, *t*-5) the effects were not significant (p > 0.34; *t*-tests). The choices one step back (*t*-1) had a stronger effect vs. those two steps back (*t*-2) ($F_{1,36}$ = 8.04, p = 0.007, $\eta_p^2$ = 0.18; mixed ANOVA). The overall effect of prior choices was

stronger in the ASD group compared to the control group ($F_{1,36}$ = 9.46, p = 0.004, $\eta_p^2$ = 0.21; mixed ANOVA). Pairwise comparisons between ASD and controls for the choice effect from one (t-1) and two (t-2) steps back (separately) found a significant difference at t-2 ($F_{1,36}$ = 12.62, p = 0.001, $\eta_p^2$ = 0.26; mixed ANOVA), with a similar trend (but not significant) at t-1 ($F_{1,36}$ = 2.65, p = 0.11, $\eta_p^2$ = 0.07; mixed ANOVA).

## Comparison to clinical scores

To test whether the increased influence of prior choices in ASD reflects clinical scores of repetitiveness and inflexibility, we correlated the $\beta_{prev\_choices}$ coefficients and ΔPSE values with the ADOS and ADI-R sub-scores (*Table 1*). There was a trend for a relationship between the overall effect of priors (ΔPSEs) and the ADI-Communication sub-scores; however, this did not survive correction for multiple (six) comparisons (p = 0.023 before correction). Correlations with the $\beta_{prev\_choices}$ coefficients were not significant (p > 0.05 before correction of multiple comparisons). A scatter of $\beta_{prev\_choices}$ vs. ADOS scores is presented in *Figure 4—figure supplement 2* ($r_{16}$ = −0.08, p = 0.77). There was also no significant correlation between $\beta_{prev\_choices}$ and SCQ ($r_{16}$ = 0.17, p = 0.49 and $r_{18}$ = −0.03, p = 0.89, for ASD and controls, respectively), nor between $\beta_{prev\_choices}$ and age ($r_{16}$ = −0.006, p = 0.98 and $r_{18}$ = −0.073, p = 0.76, for ASD and controls, respectively). A larger dataset in the future might provide more informative correlation results.

## Heading discrimination

Following the location discrimination experiment, we aimed to test whether the results would generalize to a different task, and whether it could be replicated on a different cohort. For this, we reanalyzed data from a previously published experiment that tested self-motion perception and multisensory integration of visual and vestibular cues in ASD (*Zaidel et al., 2015*). The task in this

**Table 1.** ASD participant details.

| ASD participant number | Age | SCQ | IQ – WISC IV | | | | ADOS | | | ADI-R | | |
| | | | Block-Design | Matrix-Reasoning | Vocabulary | Similarities | Calibrated Severity Score | Social Affect | RRB | Reciprocal Social Interaction | Communication | RRB |
|---|---|---|---|---|---|---|---|---|---|---|---|---|
| 1 | 8 | 17 | 7 | 9 | 10 | 9 | 9 | 7 | 7 | 13 | 11 | 5 |
| 2 | 9 | 17 | 11 | 9 | 9 | 8 | 9 | 10 | 10 | 27 | 14 | 6 |
| 3 | 10 | 21 | 13 | 16 | 13 | 13 | 8 | 8 | 8 | 10 | 13 | 1 |
| 4 | 10 | 14 | 12 | 13 | 10 | 16 | 9 | 9 | 9 | 11 | 8 | 5 |
| 5 | 10 | 11 | 9 | 7 | 7 | 7 | 9 | 9 | 9 | 12 | 12 | 2 |
| 6 | 11 | 21 | 11 | 10 | 7 | 10 | 8 | 7 | 7 | 13 | 15 | 10 |
| 7 | 11 | 15 | 7 | 11 | 8 | 7 | 8 | 8 | 8 | 11 | 11 | 5 |
| 8 | 11 | 19 | 12 | 13 | 10 | 16 | 7 | 7 | 7 | 17 | 13 | 5 |
| 9 | 11 | 22 | 12 | 9 | 11 | 10 | 7 | 10 | 10 | 23 | 19 | 7 |
| 10 | 12 | 13 | 15 | 13 | 12 | 10 | 9 | 8 | 8 | 11 | 8 | 3 |
| 11 | 12 | 29 | 10 | 12 | 7 | 8 | 7 | 7 | 7 | 20 | 19 | 4 |
| 12 | 13 | 22 | 8 | 7 | 11 | 8 | 10 | 10 | 10 | 14 | 9 | 9 |
| 13 | 13 | 12 | 7 | 10 | 8 | 10 | 10 | 9 | 9 | 23 | 11 | 4 |
| 14 | 14 | 19 | 15 | 9 | 15 | 17 | 10 | 9 | 9 | 20 | 14 | 8 |
| 15 | 14 | 14 | 14 | 9 | 12 | 15 | 7 | 5 | 5 | 11 | 7 | 5 |
| 16 | 15 | 18 | 7 | 9 | 10 | 9 | 6 | 6 | 6 | 8 | 4 | 5 |
| 17 | 16 | 30 | 13 | 9 | 11 | 12 | 8 | 6 | 6 | 13 | 13 | 8 |
| 18 | 17 | 22 | 9 | 11 | 8 | 10 | 10 | 10 | 10 | 11 | 5 | 6 |
| Mean ± SD | 12.1 ± 2.4 | 18.7 ± 5.3 | 10.7 ± 2.8 | 10.3 ± 2.3 | 9.9 ± 2.2 | 10.8 ± 3.2 | 8.4 ± 1.2 | 8.1 ± 1.6 | 8.1 ± 1.6 | 14.9 ± 5.4 | 11.4 ± 4.2 | 5.4 ± 2.3 |

SCQ scores are current from the time of the study. Other scores are from within 0–5 years of the study. RRB - Restricted and Repetitive Behaviors.

experiment was heading discrimination – also a 2AFC: participants were required to discriminate whether their self-motion was to the right or left of straight-ahead. Accordingly, the data and analyses were similar. However, the heading discrimination experiment was not originally designed to test the effects of recent priors (and therefore, there was no apriori biasing of previous stimuli). Hence, we defined priors here simply by the preceding trial.

## Larger effect of recent priors in ASD for heading discrimination

To calculate ΔPSEs, we constructed psychometric curves conditioned on the previous trial. Based on our results from the location discrimination experiment (that recent choices drive the effect), the leftward or rightward prior type for each trial was defined according to the choice on the previous trial. We first present the ΔPSE results here, and in the following section (below) we present fits to the logistic regression model.

Psychometric curves for an example ASD participant and an example control participant (visual condition) are presented in *Figure 5A* (red- and blue-shaded curves, in the left and right plots, respectively). Here too, the psychometric curves were shifted, and in the same (positive) direction as the location discrimination results: namely, following rightward choices, test stimuli were more likely to be discriminated rightward, and following leftward choices, test stimuli were more likely to be

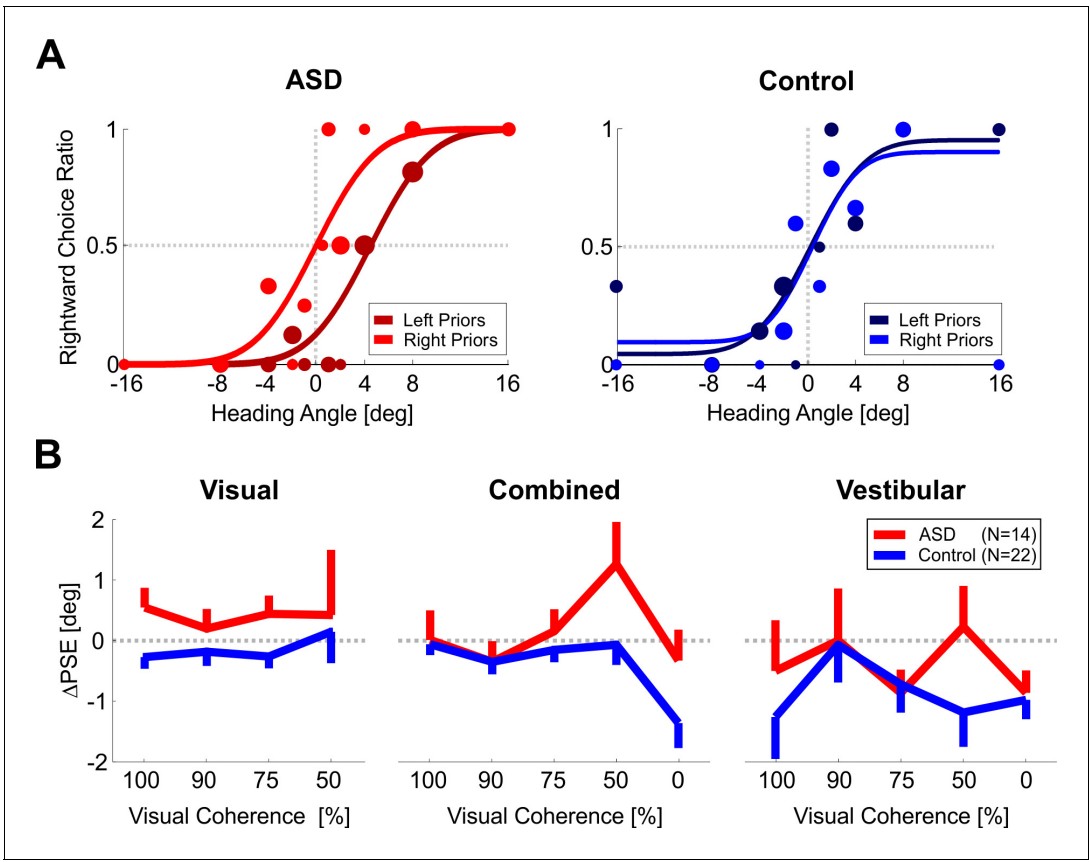

**Figure 5.** Heading discrimination – psychometric function shifts. (**A**) Psychometric curves for an example participant with ASD (left plot) and an example control participant (right plot) performing heading discrimination (visual condition, with 50% coherence). The psychometric curves represent the ratio of rightward choices as a function of heading, sorted according to the prior type (left or right bias, represented by dark and light colors, respectively). The data (circles) were fit with cumulative Gaussian distribution functions (solid lines). Circle size reflects the number of trials for a given stimulus heading and prior type. (**B**) PSE shifts (ΔPSE = PSE for left biased priors minus PSE for right biased prior) for three stimulus conditions: visual, vestibular and combined cues (left, right, and center plots, respectively). Red and blue lines represent ΔPSEs (mean ± SEM) as a function of visual coherence for participants with ASD and controls, respectively. Overall, PSE shifts are significantly higher in ASD than in controls (p = 0.040).

The online version of this article includes the following figure supplement(s) for figure 5:

**Figure supplement 1.** Stimulus convergence for the heading discrimination experiment.

discriminated leftward. Like above, in the examples here the shift for the ASD participant was larger (red colors) vs. the control participant (blue colors).

We calculated ΔPSEs for each participant, per condition: visual, vestibular, and combined visual-vestibular cues (*Figure 5B*, left, right, and center plots, respectively). Since the data comprised several blocks with different levels of visual coherence, ΔPSEs were calculated per coherence (presented along the x-axis). Also in these data, the ΔPSEs were consistently larger for the ASD group vs. controls (the red lines lie above the blue lines in *Figure 5B*). This difference was statistically significant ($F_{1,24}$ = 4.74, p = 0.040, $\eta_p^2$ = 0.17, Mixed ANOVA). These results suggest that our findings of a greater influence of prior choices in ASD generalize across multiple sensory domains. But once again, further analysis was required in order to dissociate the effects of recent stimuli and choices.

### Enhanced influence of recent choices in ASD for heading discrimination

To separate the influence of previous stimuli from previous choices, we fit the data using a logistic regression model, similar to location discrimination results presented above (*Figure 3*). But, here only one 'prior' was used (*n* = 1) because there was no consistent bias (by design) beyond the previous trial. In line with the location discrimination results, the heading discrimination model fits revealed a significant difference between ASD and control groups in the weight given to the previous choice ($\beta_{prev\_choices}$; *Figure 6*, center; $F_{1,24}$ = 4.83, p = 0.038, $\eta_p^2$ = 0.17, Mixed ANOVA). The $\beta_{prev\_choices}$ coefficients were significantly positive for the ASD group ($t_{12}$ = 3.03, p = 0.010, *Cohen's d* = 0.84, 95% CI = [0.06, 0.40], *t*-test), indicating that perceptual choices were attracted towards (i.e. more likely to be similar to) the previous choice. By contrast, the $\beta_{prev\_choices}$ coefficients were not significant in the control group ($t_{21}$ = −0.01, p = 0.99, *Cohen's d* = 0.003, 95% CI = [−0.2, 0.2], *t*-test). Thus, here too, individuals with ASD demonstrated a greater influence of prior choices.

The previous stimulus ($\beta_{prev\_stimuli}$) coefficients were significantly negative for both the ASD and control groups (*Figure 6*, right plot; $t_{13}$ = −6.73, p = 1.4•$10^{-5}$, *Cohen's d* = 1.8, 95% CI = [−0.62,−0.32] and $t_{21}$ = −6.83, p = 9.5•$10^{-7}$, *Cohen's d* = 1.46, 95% CI = [−0.51,−0.27], respectively; *t*-tests). This indicates a significant (negative) effect of adaptation to previous stimuli in the heading discrimination experiment. Accordingly, adaptation seemed to be stronger in heading discrimination vs. location discrimination, which showed a similar trend but with marginal statistics (possibly due to the

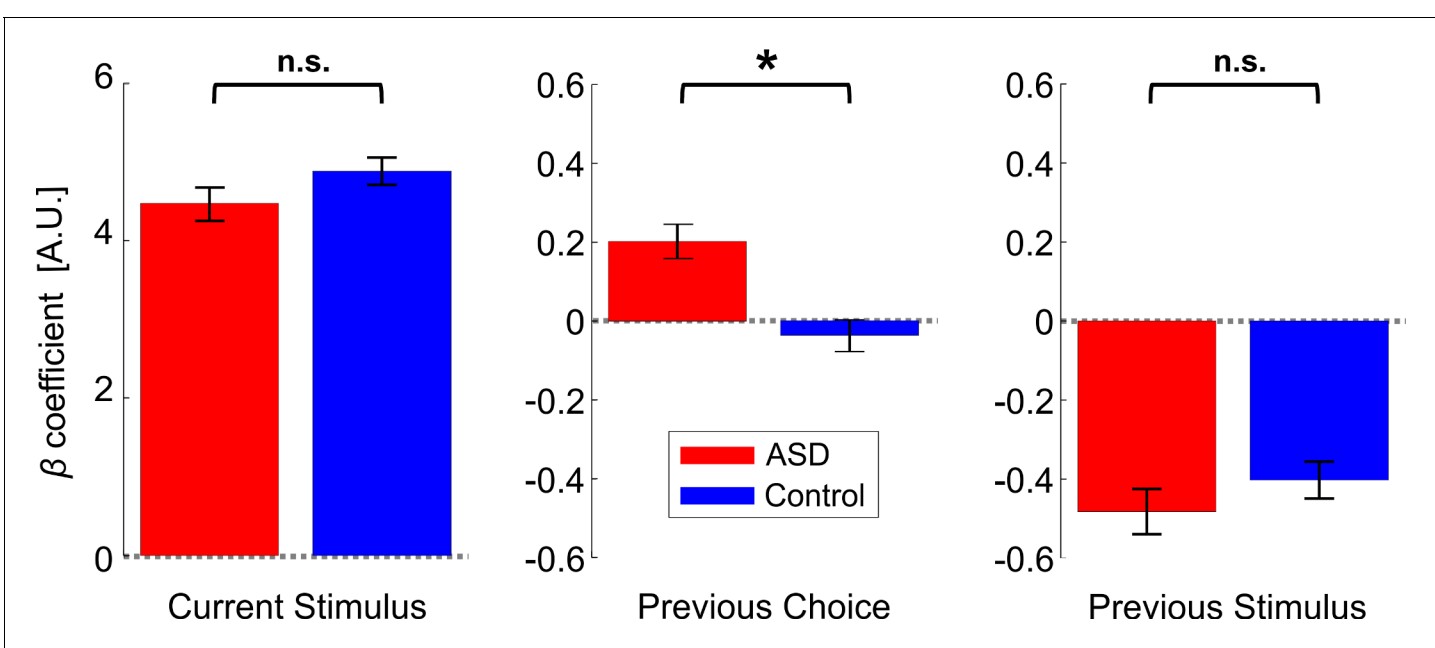

**Figure 6.** Heading discrimination – model parameters. Beta coefficients for three model parameters: current stimulus, previous choice and previous stimulus (left, center, and right plots, respectively). These were averaged over three stimulus conditions (vestibular, visual, and combined) and five visual coherence levels. Red and blue bars represent the beta coefficients (mean ± SEM) for participants with ASD and controls, respectively. *p < 0.05, n.s. – not significant.

different durations of the stimuli: 1 s for heading stimuli and 0.25 s for location stimuli). Notably, here too there was no significant difference between the $\beta_{prev\_stimuli}$ coefficients for the ASD and control groups ($F_{1,23} = 0.74$, p = 0.40, $\eta_p^2 = 0.03$, Mixed ANOVA). Thus, we found no evidence for reduced adaptation in ASD. Rather, we found an increased (positive) influence of prior choices. Since these effects can be in opposite directions, if they aren't dissociated, a greater influence of prior choices could (mistakenly) be interpreted as reduced adaptation. For example, in our vestibular data, the overall (ΔPSE) shifts lie closer to zero for ASD vs. controls (*Figure 5B*, right plot). However, this does not reflect reduced adaptation, but rather a greater influence of prior choices in ASD.

Naturally, current stimulus coefficients ($\beta_{curr\_stimulus}$) were large, positive, and significant in both ASD and control groups (*Figure 6*, left; $t_{12} = 13.35$, p = $6.8 \cdot 10^{-8}$, *Cohen's d* = 3.7, 95% CI = [3.8, 5.29] and $t_{21} = 15.7$, p = $4.46 \cdot 10^{-13}$, *Cohen's d* = 3.35, 95% CI = [4.41, 5.76], respectively; *t*-tests), indicating that, like in the location discrimination task, the current stimulus was still the most crucial factor in the perceptual decision. Therefore, the increased influence of prior choices in ASD does not simply reflect a case of blind repetition. Overall (grouping all stimulus conditions) $\beta_{curr\_stimulus}$ did not differ significantly between ASD and controls ($F_{1,24} = 0.03$, p = 0.88, $\eta_p^2 = 0.001$, Mixed ANOVA). No significant correlation was found between $\beta_{prev\_choices}$ and $\beta_{curr\_stimulus}$ ($r_{34} = 0.06$, p = 0.74). There was no significant correlation between $\beta_{prev\_choices}$ and SCQ ($r_{12} = -0.25$, p = 0.41 and $r_{20} = 0.12$, p = 0.59, for ASD and controls, respectively). Also, no significant correlation was found between $\beta_{prev\_choices}$ and age in the ASD group ($r_{12} = 0.54$, p = 0.057) or in the control group ($r_{20} = 0.15$, p = 0.50).

### Effect of recent priors in ASD ensues even when reported using different actions

In order to dissociate the effect of recent choices from motor repetition, an additional location discrimination condition was run – the *Response Invariant* condition. In this condition, different motor actions were used when responding to 'prior' vs. 'test' stimuli (identified by different colors, see Materials and methods for more details). Also in this condition ΔPSE values were significantly positive for both the control and ASD groups individually (*Figure 7A*; $t_{15} = 2.65$, p = 0.018, *Cohen's d* = 0.66, 95% CI = [0.11, 1.2] and $t_{15} = 2.68$, p = 0.017, *Cohen's d* = 0.67, 95% CI = [0.12, 1.2],

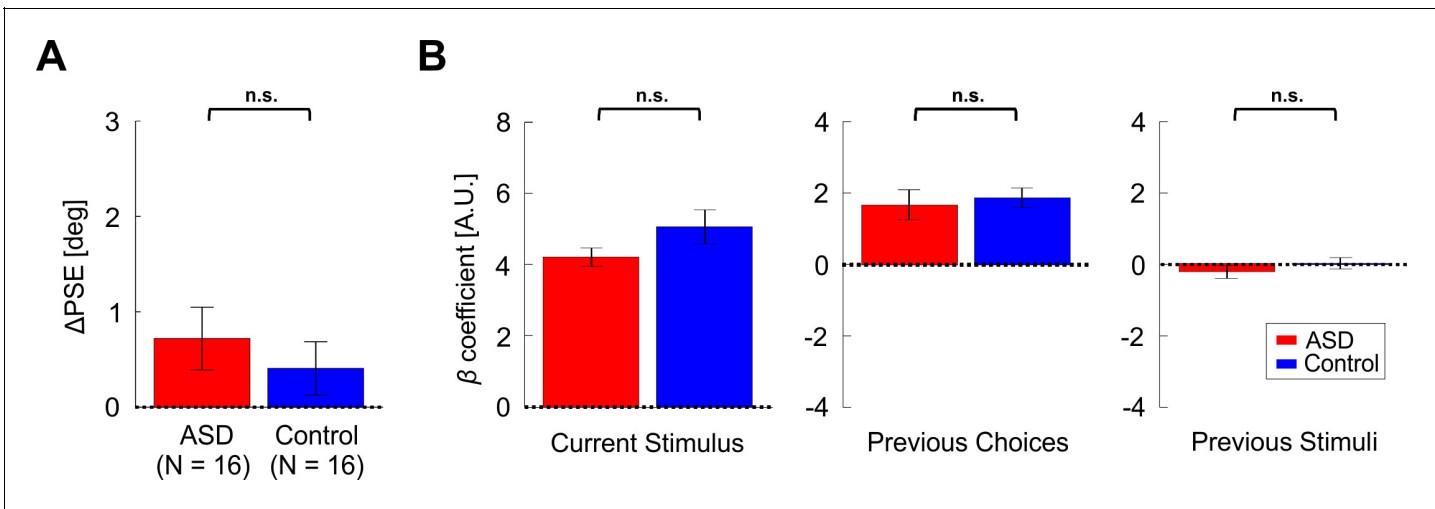

**Figure 7.** Results for the *Response Invariant* location discrimination condition. (**A**) PSE shifts (ΔPSE = PSE for left biased priors minus PSE for right biased prior). Red and blue bars represent ΔPSEs (mean ± SEM) for participants with ASD and controls, respectively. (**B**) Beta coefficients for three model parameters: current stimulus, previous choices and previous stimuli (left, center, and right plots, respectively). Red and blue bars represent the beta coefficients (mean ± SEM) for participants with ASD and controls, respectively. n.s. – not significant.

The online version of this article includes the following figure supplement(s) for figure 7:

**Figure supplement 1.** Test stimulus convergence for the location discrimination experiment, *Response Invariant* condition.

**Figure supplement 2.** Scatter of $\beta_{prev\_choices}$ coefficients (*Response Invariant* condition) vs. ADOS scores.

respectively; *t*-tests). However, there was no significant difference between the groups (*Figure 7A*; $t_{30} = 0.09$, p = 0.93, *Cohen's d* = 0.03, 95% CI = [−0.66, 0.73], *t*-test).

When comparing the logistic regression coefficients (like above for the primary condition), we found no significant difference between the groups (*Figure 7B*, $F_{3,28} = 1.18$, p = 0.34, $\eta_p^2 = 0.11$, one-way MANOVA). For both the ASD and control groups, there was a positive effect of previous choices (*Figure 7B*, center plot; $t_{15} = 3.97$, p = 0.001, *Cohen's d* = 0.99, 95% CI = [0.38, 1.58] and $t_{15} = 6.73$, p = $6.8 \cdot 10^{-6}$, *Cohen's d* = 1.68, 95% CI = [0.99, 2.44], respectively; *t*-tests) while the effect of previous stimuli was small and not significant (*Figure 7B*, right plot; $t_{15} = -1.24$, p = 0.23, *Cohen's d* = 0.3, 95% CI = [-0.8, 0.2] and $t_{15} = 0.25$, p = 0.8, *Cohen's d* = 0.06, 95% CI = [-0.43, 0.55], respectively; *t*-tests). The previous choice coefficients ($\beta_{prev\_choices}$) did not correlate significantly with ADOS scores (*Figure 7—figure supplement 2*, $r_{14} = -0.42$, p = 0.10). In line with previous findings that using the same action boosts the prior choice effect (*Feigin et al., 2021*), also here choice biases ($\beta_{prev\_choices}$) were larger in the primary condition vs. the *Response Invariant* condition ($t_{30} = 2.94$, p = 0.006, *Cohen's d* = 0.52, 95% CI = [0.15, 0.89], *t*-test; $BF_{01} = 0.15$; pooling both groups together).

Similar to the primary condition, the current stimulus effect ($\beta_{curr\_stimulus}$) was large, positive and significant for both groups (*Figure 7B*, left plot; $t_{15} = 16.26$, p = $6.2 \cdot 10^{-11}$, *Cohen's d* = 4.07, 95% CI = [2.54, 5.58] and $t_{15} = 10.74$, p = $1.9 \cdot 10^{-8}$, *Cohen's d* = 2.69, 95% CI = [1.6, 3.75], for ASD and controls, respectively; *t*-tests) and did not differ significantly between the groups ($t_{30} = -1.58$, p = 0.125, *Cohen's d* = 0.56, 95% CI = [-1.26, 0.15], *t*-test). This indicates that also in this condition sensitivity did not differ between the groups. We confirmed this by comparing thresholds from the psychometric functions, which also did not differ significantly between groups ($t_{30} = 1.02$, p = 0.32, *Cohen's d* = 0.36, 95% CI = [-0.34, 1.06], *t*-test; $BF_{01} = 2.00$). Here too, stimulus sensitivity ($\beta_{curr\_stimulus}$) did not correlate significantly with the prior choice effect $\beta_{prev\_choices}$ ($r_{30} = -0.27$, p = 0.13), and the baseline coefficients did not differ significantly between the groups ($t_{30} = 1.29$, p = 0.2, *Cohen's d* = 0.46, 95% CI = [-0.25, 1.15], *t*-test). Also, response times did not differ between the two groups: mean ± *SD* = 0.83 ± 0.15s and 0.87 ± 0.12s, for ASD and controls, respectively (calculated across all discriminations, that is, pooling prior and test discriminations; $t_{30} = -0.85$, p = 0.4, *Cohen's d* = 0.3, 95% CI = [-0.99, 0.40], *t*-test).

These results show that the positive effect of recent prior choices ensues even when context (stimulus color) and the motor actions used to report responses are altered – also for individuals with ASD. This strengthens the argument against a reduced effect of recent priors in ASD. Taking the results of the two location discrimination conditions (and heading discrimination) together, we conclude that individuals with ASD experience a positive bias from recent choices, like controls. When context is consistent (i.e. stimulus color is the same, and the same motor action is used to report choices) this effect is enlarged in ASD vs. controls. However, when context is changed (and perception-action coupling is broken) this effect is no longer enlarged in ASD (but is nonetheless still present).

## Comparable lapse rates in ASD

We hypothesized that if the increased choice effect in ASD reflects simple motor repetition, this would manifest irrespective of discrimination difficulty – that is, also for easy discriminations (ones with obvious stimuli). By contrast, an integrated effect of prior choices is expected to be negligible for the easy discriminations (i.e. because the stimuli are obvious, they are expected to dominate, masking the more subtle effect of prior choices). Thus, for a motor repetition effect, we would expect an increase in lapse rates (proportion of incorrect responses for obvious stimuli) in ASD. If, however, lapse rates are not increased in ASD, this might suggest that the increased influence of prior choices in ASD reflects a more complex decision process vs. simple motor repetition. Specifically, that prior choice information is integrated together with current stimulus information.

Lapse rates did not differ significantly between ASD and controls (raw p-values presented below without correction for multiple comparisons), with Bayes factors (slightly) favoring the null hypothesis. Lapse rates (mean ± SEM) for the location discrimination primary condition were 0.030 ± 0.008 and 0.038 ± 0.008 for ASD and controls, respectively ($t_{36} = −0.74$, p = 0.47, *Cohen's d* = 0.24, 95% CI = [−0.88, 0.40], *t*-test; $BF_{01} = 2.56$). Lapse rates (mean ± SEM) for the location discrimination *Response Invariant* condition were 0.079 ± 0.013 and 0.069 ± 0.015 for ASD and controls,

respectively ($t_{30}$ = 0.51, p = 0.61, *Cohen's d* = 0.18, 95% CI = [−0.52, 0.87], t-test; $BF_{01}$ = 2.69). Lapse rates (mean ± SEM) for the heading discrimination task were 0.015 ± 0.003 and 0.013 ± 0.003 for ASD and controls, respectively ($t_{34}$ = 0.5, p = 0.62, *Cohen's d* = 0.08, 95% CI = [−0.006 0.010], t-test; $BF_{01}$ = 2.77). Thus, no difference in lapse rates was observed in ASD, in both tasks.

To further test this point in a way that does not depend on psychometric fits and the lapse rate parameter, we also calculated and compared the percentage of correct choices for easy discriminations (defined by stimuli with absolute values larger than the 66th percentile, calculated individually for each participant and condition). Also here we found no significant differences between the ASD and control groups (raw p-values presented below without correction for multiple comparisons), with Bayes factors (slightly) favoring the null hypothesis. Percent correct (mean ± SEM) for the easy discriminations in the location discrimination primary condition was 96.61 ± 0.94% and 95.12 ± 1.16% for ASD and controls, respectively ($t_{36}$ = 0.98, p = 0.33, *Cohen's d* = 0.32, 95% CI = [−0.32, 0.96], t-test; $BF_{01}$ = 2.17). Percent correct (mean ± SEM) for the easy discriminations in the location discrimination *Response Invariant* condition was 91.21 ± 1.39% and 92.41 ± 1.75%, for ASD and controls, respectively, ($t_{30}$ = −0.54, p = 0.60, *Cohen's d* = 0.19, 95% CI = [−0.88, 0.51], t-test; $BF_{01}$ = 2.66). Percent correct (mean ± SEM) for the easy discriminations in the heading discrimination task was 88.12 ± 0.74% and 88.86 ± 0.50% for ASD and control group, respectively ($t_{34}$ = −0.87, p = 0.39, *Cohen's d* = 0.15, 95% CI = [−2.50 1.00], t-test; $BF_{01}$ = 2.28). These results suggest that the increased influence of prior choices in ASD does not reflect simple motor perseveration.

## Discussion

In this study, we found that when making perceptual decisions, individuals with ASD are more greatly influenced (vs. controls) by their recent prior choices. For all participants (ASD and control) perceptual decisions were 'attracted' toward (i.e. more likely to be the same as) recent prior choices. However, this effect was significantly larger in ASD. This finding was replicated in two different tasks (visual localization and visual/vestibular heading discrimination) performed by two different cohorts – one in Israel and one in the USA. These results are in line with a recent study that showed increased consistency in gambling choices in ASD (*Wu et al., 2018*). Here, we extend this to perceptual decision-making, and dissociated the effects of prior choices and prior stimulus information.

By contrast, the influence of recent prior sensory information was similar in ASD and controls (in both tasks). This reflected a 'repulsive' bias (away from the preceding stimuli) which was large and highly significant for both groups in the heading discrimination task (in the location discrimination task, the effect was similar, but small). Notably, this effect (adaptation to prior stimuli) was in the opposite direction vs. the effect of prior choices. A negative influence of prior stimuli on subsequent perception has been documented in many studies (e.g. sensory adaptation /aftereffects, *Gibson, 1937*; *Gibson and Radner, 1937*; *Anstis et al., 1998*; Thompson, P., and *Thompson and Burr, 2009*), even with very short (milliseconds) stimulus durations (*Sekuler and Littlejohn, 1974*; *Suzuki, 2001*; *Felsen et al., 2002*; *Xu and Liu, 2012*; *Sou and Xu, 2019*). The magnitude of adaptation aftereffects increases with exposure time (*Rhodes et al., 2007*; *Pegors et al., 2015*; *Burton et al., 2016*). This might explain why stimulus adaptation was small in the location discrimination task (0.25 s stimulus duration) and strong (and highly significant) in the heading discrimination task (1 s stimulus duration). Because there was no fixation point for the repeated stimulus presentations in the location discrimination task (all trials in the heading discrimination task had a fixation point), a negative effect of prior stimuli could reflect a shift in the perceived center of the screen.

A positive influence of prior choices has also been documented in many studies (*Kaneko and Sakai, 2015*; *Talluri et al., 2018*; *Feigin et al., 2021*). Furthermore, converging evidence from many recent studies suggests that the influence of prior stimuli and prior choices are often in opposite directions – that is, a negative (adaptive) influence of prior stimuli vs. a positive (consistency bias) influence of prior choices, and that these superimpose (*Fritsche et al., 2017*; *Bosch et al., 2020*; *Feigin et al., 2021*; *Sadil et al., 2021*). However, opposing effects of prior sensory vs. prior choices on perceptual decisions has not yet been studied in ASD. Importantly, these effects need to be dissociated in order to better understand suggestions of reduced adaptation in ASD (*Pellicano et al., 2007*; *Lawson et al., 2015*; *Molesworth et al., 2015*; *Turi et al., 2015*; *Noel et al., 2017*; *Lieder et al., 2019*).

When separating the effects of prior choices and prior stimuli, we did not find reduced adaptation to prior stimuli in ASD. Our results therefore suggest that observations of 'reduced adaptation' in ASD might (at least partially) be explained by an *increased* positive (i.e. counteracting) effect of prior choices. However, our stimuli were of relatively short duration (0.25 s in the location discrimination task and 1 s in the heading direction discrimination task). Longer stimuli might lead to greater adaptation (*Pegors et al., 2015*) which might better expose any differences. Thus, future work should test whether adaptation to longer duration stimuli (several seconds) is altered in ASD, while taking into account the effects of prior choices.

The finding of an increased influence of prior choices in ASD may suggest that repetitiveness and inflexibility symptoms might manifest also in perceptual decisions. However, we did not find a significant correlation between the beta coefficients for prior choices and ADOS and ADI-R sub-scores of repetitiveness and inflexibility. Accordingly, our findings might offer an additional measure of repetitive symptoms, which is perhaps not covered by these sub-scores (but this would need a larger data set to confirm). Although there was a trend between the overall effect of priors (ΔPSEs) and the ADI-Communication sub-scores, this did not survive correction for multiple comparisons. Therefore, whether or not an increased influence of prior choices correlates with ASD symptoms warrants further investigation with a larger dataset in the future.

The *Response Invariant* condition tested the effect of prior choices when context was altered – choices were reported via different motor actions (signaled by different colored stimuli). Also here, individuals with ASD still showed a significant influence of prior choices. This challenges the theories of attenuated priors in ASD (*Pellicano and Burr, 2012*) and demonstrates that even when context is different, individuals with ASD nonetheless still demonstrate a robust influence of priors (specifically prior choices). However, in this condition, the effect was no longer larger than controls. This could indicate that the primary results reflect 'motor perseveration' in ASD rather than a greater influence of prior choices. Although we cannot rule out a contribution of motor perseveration, several additional analyses suggest that it does not fully account for the ASD results. Firstly, a simple motor perseveration effect would manifest irrespective of discrimination difficulty (also for easy discriminations), whereas we found no significant increase in lapse rates in ASD. Also, we found an independent contribution of choices before the previous choice (i.e. two choices back; with significantly larger effects in ASD). This further suggests that the increased effect of prior choices in ASD reflects a compound influence of choices, beyond simple motor repetition.

The observation that an enlarged influence of prior choices in ASD was seen only in the primary condition (and in the heading discrimination condition), when motor responses to prior and test stimuli were the same, but not when the actions for prior and test stimuli were dissociated, suggests that this difference in ASD may be mediated by altered perception-action coupling. Namely, the greater influence of prior choices in ASD may be specific to when perception-action coupling is consistent. This interpretation is in line with the hypothesis that perception and action are intertwined in a perception-action loop (*Buckingham et al., 2016*) and our recent finding (in a series of related control experiments; *Feigin et al., 2021*) that sensorimotor (decision-action) coupling boosts the influence of prior choices. Accordingly, the increased influence of prior choices in ASD may reflect an increased influence of prior perception-action coupling on subsequent decisions.

The consistency bias in the heading discrimination task was less strong (and not seen for control participants). We think that this might be because of: (i) relatively long inter-stimulus-intervals (5~6 s, due to technical constraints – the motion platform needs to return slowly to base before a new trial can begin). By contrast, ISIs in the location discrimination task were ~1 s. (ii) The heading discrimination task involved three types of stimuli from two different sensory modalities: visual, vestibular, and combined (both visual and vestibular stimuli delivered together). The stimuli were interleaved, so that in most trials, the current and prior stimulus were of different types. The effect of the previous trial may be decreased when the conditions differ across trials (*Feigin et al., 2021*).

It has been proposed that individuals with high-functioning ASD have an increased ability and tendency to recognize repeating patterns in stimuli and to generate rules for prediction (*Baron-Cohen, 2002*; *Baron-Cohen, 2009*). Our results suggest that an elaborate set of systematic, internal rules in ASD uses information from prior choices to a greater degree. Furthermore, consistency (in context and in the motor responses) might be an important factor for generating prediction rules in ASD, which has been proposed to be a disorder of prediction (*Sinha et al., 2014*).

In summary, we found here that individuals with ASD demonstrated an increased influence of recent prior choices on perceptual decisions (vs. controls), while the influence of recent stimulus history was unaltered. This was seen in two different tasks using different modalities (visual localization and visual/vestibular heading discrimination), tested in two different cohorts (Israel and USA). These results suggest that perceptual decisions in ASD are more repetitive and less flexible (i.e. more consistent with prior decisions).

## Materials and methods

We studied serial dependence of perceptual decisions in individuals with ASD vs. controls performing two different tasks: (i) visual location discrimination (designed for this study and performed in Israel), (ii) multisensory (visual-vestibular) heading discrimination (reanalysis of previously published data, collected in the USA; *Zaidel et al., 2015*). We present below participant and task details of the former (visual location discrimination), and briefly summarize relevant details of the latter (heading discrimination), for which full details are available in the original publication.

### Participants

The visual location discrimination task was performed by 23 children and adolescents with ASD and 27 age-matched controls (all male; age range: 8–17 years). This study was approved by the institutional Helsinki committee at The Shamir Medical Center (0214–15-ASF) and the internal review board at Bar-Ilan University. This sample size was determined based on the results of our previous study which tested the same paradigm in young adults (*Feigin et al., 2021*). Although performance in children may differ, the task was identical, and therefore this offered the best available estimate. All participants (and one of their caregivers) signed informed consent. Participants with ASD were recruited through The Autism Center/ALUT at The Shamir Medical Center in Israel, and were all high functioning and cooperative. The ASD group was selected from a population that underwent a comprehensive evaluation for (and diagnosis of) ASD at a tertiary autism center. The criteria for selection of the ASD participants included having verbal and nonverbal scores equal to or above 80 points on the Wechsler Intelligence Scale for Children – Fourth Edition (WISC-IV; *Wechsler, 2010*).

The behavioral assessment of ASD was based on two standardized tests, the Autism Diagnostic Interview-Revised ADI-R (*Lord et al., 1994*; *Rutter et al., 2003b*) and the Autism Diagnostic Observation Schedule (ADOS-II; *Lord et al., 2012*), and a clinical judgement based on the DSM-IV criteria (*American Psychiatric Association, 2000*). All participants (including controls) were screened at enrollment with the Social Communication Questionnaire (SCQ) current version (*Rutter et al., 2003a*).

To verify that the ASD and control groups were matched in terms of intellectual functioning, the control group was tested using two verbal (vocabulary, similarities) and two nonverbal (block-design, matrix-reasoning) WISC-IV subtests (these scores were available for the ASD group from previous assessment). Performance of the ASD and control groups for the four subtests did not differ significantly (p > 0.12 for all four pairwise comparisons). Therefore, there is no reason to believe that any task performance differences between the groups resulted from differences in IQ. All participants had normal or corrected-to-normal vision.

One of the participants from the ASD group and two participants from the control group did not complete the task and were therefore excluded. The data of the remaining participants were pre-screened to confirm task understanding and cooperation (see *Data Analysis – psychometric fits* section below for details). This removed an additional four ASD participants and five control participants, leaving 18 participants in the ASD group and 20 participants in the control group for further analysis. *Tables 1* and *2* present the ages, IQ and SCQ scores for the ASD and control groups, respectively, with clinical scores (for the ASD group) also presented in *Table 1*.

The heading discrimination experiment tested 14 adolescents with ASD and 22 age-matched controls (all male; ages 13–19). It was approved by the institutional review board at Baylor College of Medicine, where the study was performed. Further details regarding the cohort and task are available in the original publication (*Zaidel et al., 2015*).

**Table 2.** Control participant details.

| Control participant number | Age | SCQ | IQ – WISC IV | | | |
|---|---|---|---|---|---|---|
| | | | Block-Design | Matrix-Reasoning | Vocabulary | Similarities |
| 1 | 8 | 5 | 14 | 10 | 18 | 12 |
| 2 | 9 | 1 | 14 | 10 | 15 | 10 |
| 3 | 9 | 1 | 13 | 13 | 17 | 14 |
| 4 | 9 | 1 | 13 | 14 | 14 | 15 |
| 5 | 10 | 1 | 11 | 10 | 10 | 12 |
| 6 | 11 | 1 | 9 | 15 | 15 | 8 |
| 7 | 11 | 3 | 10 | 8 | 9 | 10 |
| 8 | 12 | 10 | 6 | 11 | 10 | 9 |
| 9 | 12 | 1 | 12 | 7 | 8 | 9 |
| 10 | 12 | 1 | 9 | 8 | 11 | 9 |
| 11 | 13 | 3 | 14 | 12 | 13 | 18 |
| 12 | 13 | 0 | 5 | 8 | 8 | 9 |
| 13 | 13 | 8 | 14 | 17 | 11 | 16 |
| 14 | 14 | 3 | 10 | 15 | 8 | 13 |
| 15 | 15 | 0 | 10 | 14 | 12 | 13 |
| 16 | 15 | 5 | 12 | 16 | 9 | 7 |
| 17 | 15 | 7 | 13 | 16 | 12 | 15 |
| 18 | 16 | 2 | 14 | 14 | 8 | 11 |
| 19 | 16 | 3 | 8 | 8 | 9 | 9 |
| 20 | 16 | 2 | 11 | 10 | 5 | 11 |
| Mean ± SD | 12.5 ± 2.6 | 2.9 ± 2.8 | 11.1 ± 2.7 | 11.8 ± 3.2 | 11.1 ± 3.4 | 11.5 ± 3.0 |

IQ and SCQ scores are current from the time of the study.

## Location discrimination task - primary condition

The location discrimination experiments were run in a dark, quiet room at the Gonda Multidisciplinary Brain Research Center, at Bar-Ilan University. Participants were seated comfortably in front of an LCD monitor (resolution 1920×1200 pixels, refresh rate 60 Hz) and their head position fixed by a chin-rest. The screen was positioned such that the center of the screen lay 50 cm directly in front of the participant's eyes. The stimuli were filled circles (6.36° visual angle diameter), generated and presented using PsychoPy software (*Peirce, 2007*) on a dark gray background. The luminance of the circles was 161.4 cd/m$^2$ and the luminance of the background was 31.7 cd/m$^2$. Circle locations were fixed in the vertical plane (halfway up the screen) and varied in the horizontal plane only (right/left). At the screen center the stimulus ($s$) was defined as $s = 0°$ such that circle locations positioned to the right and left of the screen center had positive ($s > 0°$) and negative ($s < 0°$) values, respectively. As the magnitude of $s$ decreases (approaches zero), the stimulus becomes more difficult to discriminate.

The participants performed a two-alternative forced choice (2AFC) task in which they were required to discriminate the location of the circle stimulus (left or right relative to the screen's center) by pressing the corresponding arrow key (right or left arrow key, respectively) on a standard computer keyboard. They were instructed to make their choice as accurately and as quickly as possible. They were informed that task difficulty varies (according to the distance of the circle from the screen center) and instructed to make their best guess when in doubt.

The stimulus locations followed a structured and balanced paradigm that we have recently developed to induce and test short-term biases over several (prior) stimuli (*Feigin et al., 2021*) as follows: each trial comprised of a series of circle stimuli, starting with a random number of biased 'prior' circles, followed by a single (unbiased) 'test' stimulus. The number of prior circles ($n$) was drawn from a discrete normal distribution of natural numbers limited to the range 1–9 (with mean $\mu = 5$, and

standard deviation $\sigma$ = 2). On a given trial, the locations of the prior circles were drawn randomly from a normal distribution, biased either to the left ($\mu$ = −3.82, $\sigma$ = 3.82°) or to the right ($\mu$ = 3.82, $\sigma$ = 3.82°). Trials with leftward or rightward biased priors were randomly interleaved. The priors on a given trial were biased in one direction (on average), such that they could still appear to the contralateral side, but with lower probability (~16%).

Test circle locations were balanced to the left/right, with the location sign randomly selected on each trial ($p$ = 0.5 positive/negative). Test location magnitude (absolute distance from 0°) was set using a staircase procedure (*Cornsweet, 1962*). We considered that a staircase procedure would be preferred for this experiment because it increases the amount of information attained for the same number of trials. It does this by focusing on the difficult trials (which are informative because of the mistakes that participants make) and 'wasting' less trials with easy stimuli (for which participants often attain close to perfect performance, and are thus less informative). The staircase parameters used here were the same as our recent study using the same paradigm in adults (*Feigin et al., 2021*).

The starting location magnitude was $|s|$ = 6.36° and followed a 'one-up, two-down' staircase rule: after a single incorrect response, the location magnitude $|s|$ increased (i.e. became easier), and after two successive correct responses, the magnitude decreased (i.e. became harder). We chose this staircase ratio in order to have enough difficult trials. We reasoned that these would be more informative in measuring the influence of prior trials, which might be less observable for obvious stimuli. We confirmed that the staircases converged around the expected ~70.7% correct performance (*Leek, 2001*). Because stimulus laterality (right/left) was counterbalanced, a slight increase in the number of easy trials that could result from a bias is not expected to affect the biases or thresholds measured (in the same way that these could also be measured, albeit less efficiently, using the method of constant stimuli).

Staircase step size followed logarithmic increments, where location magnitude $|s|$ was multiplied by 0.6 to increase task difficulty and divided by 0.6 to decrease task difficulty. This step size was chosen (rather than smaller increments) because quick convergence was important with a limited number of staircase (test) trials (prior discriminations were not part of the staircase). Location magnitude was limited to a maximum of $|s|$ = 25.46°, but this was never reached in practice. Separate staircase procedures were run for each prior type (rightward/leftward biased, each comprising 100 trials) and pseudo randomly interleaved. The test stimulus magnitude for the location discrimination task as a function of trial number (the two staircases interleaved) is presented in *Figure 2—figure supplement 1* (primary condition) and *Figure 7—figure supplement 1* (*Response Invariant* condition). Staircase convergence for heading discrimination (vestibular and visual trials) is presented in *Figure 5—figure supplement 1*.

Using this paradigm, we were able to examine the short-term effects of the priors' bias on subsequent perceptual decisions, in a controlled and interleaved manner. The event sequence within a trial (and across trials) is presented in *Figure 1*. At the beginning of each trial, a fixation cross was presented at the center of the screen for 500 ms, with the intention of reducing possible carry-over effects from the previous trial. The fixation cross was not presented again during the rest of the trial in order not to interfere with the prior's effect being measured. Each circle (prior/test) was presented for 250 ms, and was preceded by a 250 ms delay (blank screen) such that the next circle appeared 250 ms after the participant's response (the stimulus was delayed until participant's response, which could be slower or faster). Testing both prior types (right and left, 200 trials together) took ~20 min. A break was automatically given after 100 trials, and the experiment was resumed by the participant, when ready.

## Location discrimination - *Response Invariant* condition

Thirty-two participants (16 ASD and 16 controls) from those who partook in the primary location discrimination experiment (described above) agreed to perform a second location discrimination task – the *Response Invariant* condition, right after the first (a break was given in between the two tasks). This condition was designed to dissociate the effect of previous choices from previous motor responses on subsequent perceptual decisions. The stimuli and procedures were the same as before, except: (i) prior circles were colored (either green or purple) while test circles remained white (all priors and test stimuli had equal brightness). (ii) The choices for prior (colored) circles were reported using a different set of keys from the test (white) circles. Participants were instructed to use the 'up'

and 'down' arrow keys to report right and left choices for colored circles (counterbalanced across participants), and the 'right' and 'left' arrow keys for white circles (like before). The color was randomly chosen for each prior stimulus (50% green and 50% purple). Only distinction between colored vs. white circles was required (no distinction was needed between purple and green circles), but we still used two colors in order to be consistent with our previous study in which different colors were used because of other manipulations (*Feigin et al., 2021*).

## Heading discrimination task

In order to investigate whether the effects of recent biased priors, revealed by our location discrimination task, generalize across other tasks, we reanalyzed data from a previous study of heading discrimination in ASD. Extended details of this experiment are found in the original publication (*Zaidel et al., 2015*). In brief: participants were seated on a motion platform, and viewed a screen (also mounted on the platform) through 3D glasses. They were presented with self-motion stimuli that were either vestibular-only (inertial motion of the motion platform, in darkness), visual-only (optic-flow simulating self-motion of the participant through a cloud of 'stars', while the motion platform remained stationary), or combined vestibular and visual cues (inertial motion in conjunction with synchronized optic flow). The self-motion stimuli were primarily in a forward heading direction (single-interval), with slight deviations to the right or left of straight ahead. The participants' task was to discriminate whether the heading direction was to the right or left of straight ahead (2AFC).

Visual cue reliability was varied by manipulating the motion coherence of the optic-flow pattern (the percentage of stars moving coherently). Vestibular reliability was fixed throughout the experiment. Five levels of visual coherence were tested, in separate blocks: 100%, 90%, 75%, 50%, and 0%. One coherence was tested per block. Visual, vestibular, and combined cue conditions were interleaved, each with their own staircase. For 0% coherence the visual only condition was not tested (only vestibular and combined cue conditions were tested) because 0% coherence represents complete visual noise.

This experiment was not originally designed to assess the effects of priors, and thus test stimuli were not preceded by a series of biased priors. So, for these data each trial was analyzed as a 'test' stimulus, conditioned on the previous trial (which served as its 'prior'). To analyze psychometric shifts due to priors (see below), the trials were sorted based on the choice of the previous trial. In the subsequent model analysis (see below), the effects of both prior stimuli and choices were accounted for.

## Data analysis – psychometric fits

Data analysis was similar for the location discrimination and heading discrimination data, and performed with custom software using Matlab (The MathWorks). Psychometric functions were defined by the proportion of rightward choices as a function of stimulus $s$ (circle location or heading direction) with positive and negative values representing stimuli to the right and left, respectively, and $s = 0°$ marking straight ahead (screen center for location discrimination). Psychometric functions were calculated by fitting the data (per participant, condition) with the following function:

$$\varphi(s) = \lambda + (1 - 2\lambda)F(s; \mu, \sigma) \tag{1}$$

where *F(s)* is a cumulative Gaussian distribution (with mean μ and standard deviation $\sigma$) and $\lambda$ represents the lapse rate. The mean (μ) reflects the point of subjective equality (PSE; the stimulus level with equal probability for a right/left choice), and we define the psychophysical threshold by the standard deviation ($\sigma$). Smaller thresholds (smaller $\sigma$) reflect better performance.

The goodness-of-fit of the psychometric functions was evaluated using the Likelihood-ratio based *pseudo-R-squared* (*Hosmer and Lemeshow, 1989*; *Menard, 2000*), which was calculated by the proportional reduction in the deviance of the fitted psychometric model ($D_{fitted}$) compared to that of the null model ($D_{null}$):

$$R_L^2 = 1 - \frac{D_{fitted}}{D_{null}} \tag{2}$$

Only participants whose psychometric curves all produced $R_L^2 > 0.5$ were included the analysis. Psychometric functions and deviances were calculated using the *psignifit* toolbox for Matlab (Version 4) (*Schütt et al., 2016*).

The aggregate effect of priors on subsequent discriminations was assessed by calculating separate psychometric functions for the test stimuli preceded by left and right biased priors and calculating the difference in PSE, as follows:

$$\Delta PSE = PSE_{left} - PSE_{right} \tag{3}$$

where the sign of ΔPSE corresponds the direction of the priors' effect. Positive ΔPSE values indicate an attractive effect (perceptual choices are more likely to be to the same side as the prior bias) and negative values indicate a repulsive effect (perceptual choices are more likely to be to the opposite side of the prior bias).

Lapse rates were extracted by fitting a psychometric curve (with a lapse rate parameter, $\lambda$) to each participant's data (pooling all test and prior discriminations, irrespective of prior type), per condition. To further test lapses in a way that does not depend on psychometric fits and the lapse rate parameter, we also calculated and compared the percentage of correct choices for easy discriminations (defined by stimuli with absolute values larger than the 66th percentile, calculated individually for each participant and condition). For the heading discrimination data, lapse rates were calculated for each stimulus type separately, and then averaged (per participant) for comparison between the ASD and control groups.

We tested for differences in performance/sensitivity between the ASD and control groups by comparing the thresholds ($\sigma$) extracted from the psychometric fits. To reduce unwanted effects of 'priors' on the threshold estimates, psychometric curves were fit separately, per prior type, and the thresholds averaged (per participant). Because thresholds are non-negative and scale logarithmically, averaging and statistical comparisons were performed on the natural log values.

## Perceptual decision model fit

While the measure of ΔPSE (*Equation 3*) quantifies the aggregate influence of the prior discriminations, it does not dissociate the effect of the prior stimuli from that of the prior choices. Moreover, these can have opposing effects and therefore (at least partially) cancel each other out. To address this, we fit a binomial logistic regression model (per participant and condition) that predicts a participant's choices based on four factors: (i) the current stimulus, (ii) previous stimuli, (iii) previous choices, and (iv) the participant's baseline bias to choose right or left (see *Figure 3* for a schematic of the model). Specifically, the logistic regression model estimates the probability of making a rightward choice ($p_t$) for the current stimulus (at time *t*) as follows:

$$p_t = \frac{1}{1 + e^{-z_t}} \tag{4}$$

where $z_t$ is a linear combination of the factors:

$$z_t = \beta_0 + \beta_{curr\_stimulus} \cdot s_t + \beta_{prev\_stimuli} \cdot \frac{1}{n}\sum_{i=t-n}^{t-1} s_i + \beta_{prev\_choices} \cdot \frac{1}{n}\sum_{i=t-n}^{t-1} c_i \tag{5}$$

Here, $\beta_0$ represents the participant's baseline bias, $s_t$ is the current stimulus (at time *t*) and $\beta_{curr\_stimulus}$ its fitted weight. The other two terms account for the effects of previous stimuli and choices. Namely, $\frac{1}{n}\sum_{i=t-n}^{t-1} s_i$ is the average stimulus from the *n* previous discriminations (and $\beta_{prev\_stimuli}$ its fitted weight) and $\frac{1}{n}\sum_{i=t-n}^{t-1} c_i$ is the rightward choice proportion from the *n* previous discriminations (and $\beta_{prev\_choices}$ its fitted weight). For the location discrimination data, responses to the test stimuli were fit using all the *n* priors (per trial). For the heading discrimination data only the previous discrimination was used as a prior (*n* = 1).

Choices were binary (right/left; coded by '1' and '−1', respectively), while stimulus values (circle locations or heading directions) were continuous (with negative and positive values for stimuli to the left/right, respectively). In order to allow for better comparison between the logistic regression weights, stimulus intensities were normalized (before model fitting) by the root-mean-square (RMS) of all the actual stimulus values presented in the task. That way, stimulus and choice factors were of comparable magnitude, both with RMS = 1.

Although correlated, prior choices and prior stimuli were separable in the model for two reasons: (i) choices are binary, whereas the stimuli are graded, and (ii) participants make mistakes. Regarding

the first point, we did not test/model (another) binary parameter of stimulus laterality, because that would imply it has undergone classification (making it 'choice-like', and thus difficult to interpret). Rather, we reasoned that a graded stimulus parameter is more reflective of the stimulus (adaptation) effects we aimed to model – in line with traditional sensory adaptation literature that shows effects proportional to stimulus strength (*Bosch et al., 2020*), rather than to binary stimulus categories. Overall, Pearson correlations between stimuli and choices in the location discrimination tasks (across all discriminations in the block) ranged on average 0.61–0.70 (depending on condition), that is, $R^2 \leq$ 0.5. Accordingly, stimulus and choice parameters contained unique information to be fit by the model.

To investigate whether there was a differential contribution of 'priors', we performed an additional regression analysis, designed to study the separate effects of five preceding discriminations. This was done by expanding Equation 5 to have five parameters (for *t*-1 to *t*-5), instead of one, for $\beta_{prev\_stimuli}$ and $\beta_{prev\_choices}$. Because the number of 'priors' for a given trial in the location discrimination task could be less than 5 (range 1–9), we fit the model to all the data (prior and test discriminations, viewed as one long sequence of trials). Namely, for each discrimination we considered the five preceding discriminations as 'priors' (irrespective of whether they were initially termed 'prior' or 'test' stimuli). This was only possible in the primary location discrimination condition because 'prior' and 'test' circles did not differ in terms of stimulus color and the motor actions used to report choices. To prevent overfitting and spurious results that could arise from too many parameters, we first investigated whether the effect of prior choices and prior stimuli were significant for the most recent two prior discriminations. Only the prior choices were significant (see Results), hence we investigated further only the effects of the prior choices, 5-back (seven-parameter model, including $\beta_{curr\_stimulus}$ and $\beta_0$).

## Statistics

Statistics analyses were performed using JASP (version 0.8.6.0), SPSS Statistics for Windows (version 21.0) and Matlab. Our apriori hypothesis for this study was that in ASD perceptual decisions would be influenced differently by prior discriminations. Two separate location discrimination datasets were gathered for this study (the primary condition and the *Response Invariant* condition). In addition, an old dataset (heading discrimination) was used to test replicability, and generalization of the results. For each dataset, we tested whether ΔPSE (*Equation 2*) was significantly different from zero, using a two-tailed *t*-test. Since these were different datasets (each with the same hypothesis), we did not adjust the p-values when assessing ΔPSE.

Our model analysis specifically aimed to compare the separate influences of previous choices and previous stimuli (two parameters: $\beta_{prev\_choices}$ and $\beta_{prev\_stimuli}$) between ASD and controls. To correct for this, we multiplied the *p*-values for these comparisons by two (Bonferroni correction) for significance testing. When reporting significant results we report the p-values after multiplying them by two and we present 97.5% (in lieu of 95%) confidence intervals (CIs). For non-significant results, we report the raw (uncorrected) p-values. The other two model parameters ($\beta_0$ and $\beta_{curr\_stimulus}$) were needed for the model fits (to better estimate the parameters of interest). They were only compared statistically for control and sanity checks (e.g. to confirm a strong influence of the current stimulus) not for hypothesis testing – hence their p-values were not corrected. When reporting one-sample *t*-tests within a group, we report the raw p-values without correction. The broader model analysis of five back priors (location discrimination primary condition) was exploratory in nature, with no correction.

The heading discrimination data analysis was applied in order to confirm and replicate our findings from the location discrimination experiment. For this analysis, we specifically hypothesized a difference in the effect of previous choices in ASD (therefore, no correction was applied). Although the hypothesized effect was in a specific direction (i.e. a greater influence of prior choices in ASD), which could justify using a one-tailed *t*-test, we used a two-tailed test (which is stricter) for better statistical robustness.

## Data and code

The data and analysis code for the location discrimination experiments (i.e., the new data from this study) can be found at https://github.com/HF-GH/ASD (copy archived at swh:1:rev:

313acbcb7ec08606d1a3332c6ee8645a091aae1a, *Feigin, 2021*). A function to plot and view the individual psychometric curves for each participant (for both the primary and the *Response Invariant* conditions, 70 sessions × two psychometric plots) can also be found there.

## Acknowledgements

We thank Mor Weinberger and Yarden Menashri-Sinai for help with location discrimination data collection, and Tamar Harpaz for administrative assistance. This work was supported by grants from The Israeli Centers of Research Excellence (I-CORE, Center No. 51/11) and from the Israel Science Foundation (ISF, grant No. 1291/20) to AZ.

## Additional information

### Funding

| Funder | Grant reference number | Author |
| --- | --- | --- |
| Israeli Centers for Research Excellence | Center No. 51/11 | Adam Zaidel |
| Israel Science Foundation | Grant No. 1291/20 | Adam Zaidel |

The funders had no role in study design, data collection and interpretation, or the decision to submit the work for publication.

### Author contributions

Helen Feigin, Data curation, Software, Formal analysis, Validation, Investigation, Visualization, Methodology, Writing - original draft; Shir Shalom-Sperber, Software, Formal analysis, Validation, Investigation, Visualization, Methodology, Writing - original draft; Ditza A Zachor, Resources, Data curation, Writing - review and editing; Adam Zaidel, Conceptualization, Resources, Data curation, Formal analysis, Supervision, Funding acquisition, Validation, Investigation, Methodology, Writing - original draft, Project administration, Writing - review and editing

### Author ORCIDs

Helen Feigin (iD) https://orcid.org/0000-0002-6069-0416
Shir Shalom-Sperber (iD) https://orcid.org/0000-0003-2455-5871
Adam Zaidel (iD) https://orcid.org/0000-0003-4405-8717

### Ethics

Human subjects: This study was approved by the institutional Helsinki committee at The Shamir Medical Center (0214-15-ASF) and the internal review board at Bar-Ilan University. All participants (and one of their caregivers) signed informed consent.

### Decision letter and Author response

Decision letter https://doi.org/10.7554/eLife.61595.sa1
Author response https://doi.org/10.7554/eLife.61595.sa2

## Additional files

### Supplementary files

• Transparent reporting form

### Data availability

The data and analysis code for the location discrimination experiments (i.e., the new data from this study) have been uploaded to github and can be found at https://github.com/HF-GH/ASD (copy

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
