## [Decision Letter]

**Acceptance summary:**

The study addresses a timely and important question of the role of prior choices on perceptual decisions in high functioning children and teenagers diagnosed with autism (Autism Spectrum Disorder). The experiments are well motivated and thoughtfully designed. Using a model to dissociate the contribution of prior stimuli and choices, the authors found a strong effect of prior choices not stimuli, which is stronger in ASD than controls. Similar results from another data set are also reported.

**Decision letter after peer review:**

Thank you for submitting your article "Increased influence of prior choices on perceptual decisions in autism" for consideration by *eLife*. Your article has been reviewed by 3 peer reviewers, and the evaluation has been overseen by a Reviewing Editor and Joshua Gold as the Senior Editor. The following individual involved in review of your submission has agreed to reveal their identity: Yoram Boneh (Reviewer #3).

The reviewers have discussed the reviews with one another and the Reviewing Editor has drafted this decision to help you prepare a revised submission.

Summary:

The study addresses a timely and important question of the role of prior choices on perceptual decisions in individuals diagnosed with autism; 17 high functioning (but not mild cases) children and teenagers (8-17 years) with ASD. The experiments are well motivated and thoughtfully designed. Using a model to dissociate the contribution of prior stimuli and choices, the authors found a strong effect of prior choices not stimuli, which is stronger in ASD than controls. Similar results from another data set are also reported.

Overall, this is an strong study with a sophisticated protocol, elaborate data analysis, ASD participants who were tested on a large battery, in-depth analysis of the literature with interesting insights, interesting results and a well written manuscript.

The first two experiments provided compelling evidence that prior choices affect perceptual decision making in ASD, but the outcome of the response invariant condition suggests that the authors' interpretation goes beyond the data. This has serious implications for the interpretations of the findings. Also, the bias interpretation should be informed by measures of performance.

Essential revisions:

1. "In summary, we found here that individuals with ASD demonstrated an increased influence of recent prior choices on perceptual decisions (vs. controls),…" is the major finding, quoted here in the concluding paragraph. It seems, however that the data support a narrower (and potentially less interesting) conclusion that individuals with ASD demonstrated an increased influence of recent button presses/motor responses, as the finding which forms the basis of the summary went away when different keys were used to report prior vs. test responses (i.e., in the response invariant condition). We understand that the authors present these data as a challenges to theories of attenuated priors in ASD, but they seem to sidestep the issue that these data make their general conclusion more complicated and weaker. If there was "Increased influence of prior choices on perceptual decisions in autism", as opposed to increased influence of prior motor actions, then the group difference in the response invariant condition should have been replicated.

2. Related to the previous point, the word "perseveration", i.e. a tendency to repeat the last key or recent keys is not mentioned. The authors conducted a "response invariant" experiment which showed significant but much smaller biases (Figure 7). Are these significantly smaller than the 1st experiment (as seems to be the case from the plots)? If so, one cannot rule out a major contribution of repeating the recent keys, i.e. perseveration. It would be useful to see the raw data in this case; e.g. what is the %trials of pressing right when the priors were biased to the right. Likely this was high, given that the staircase was symmetric (50/50 trials on left and right) and that a bias emerged from the data.

3. The finding from the model that the prior stimuli did not have a positive impact (and even negative) on the decision bias is counter-intuitive and needs explanation. There were typically 5 prior trials, ~4 of them on one side, e.g. right, resulting in a higher rate of right presses on the test (because the test was unbiased, and the results showed a bias). Assuming the prior trials were mostly replied correctly, there should be a correlation between the stimuli and the choices. There are two possible reasons why the model produced negative weights – one is that indeed the choices were different from the stimuli, in which case we need to know the performance of the participants on the prior trials (which would be useful anyway). The other possibility is that the choices for the model were binary and the stimuli were continuous. If the stimuli had been coded as binary, it would have been difficult to dissociate between the stimuli and the choices. In this case, the conclusion should be that the prior stimulus laterality could have impacted the test choices, but not their magnitude. This issue should be explained in the text.

4. Currently, all previous trials are treated equal in the model, but their contribution is not necessarily equal. Could you analyze the data to assess whether there is a differential contribution of preceding trials, i.e. the details of the serial dependency.

5. The performance on the test trials staircase procedure is not reported, only the PSE difference. Thresholds and slopes, indicators of sensitivity and precision of performance in the task are not detailed. Bias magnitude could vary as a factor of noise or sensitivity.

a. Biases are likely to push the staircase procedure to higher laterality discrimination thresholds. Could worse performance (more errors) on the staircase procedure amplify (but not create) the bias?

b. Was sensitivity considered in anyway in the analysis?

c. Did performance differ between groups? Example psychometric curves shown seem shallower in ASD.

d. Was performance correlated with bias?

e. Could larger consistency bias in the ASD group result from weaker performance, more lapses of attention?

f. How would the model fitting look like and how would it interact with the biases?

---

## [Author Response]

Essential revisions:1. "In summary, we found here that individuals with ASD demonstrated an increased influence of recent prior choices on perceptual decisions (vs. controls),…" is the major finding, quoted here in the concluding paragraph. It seems, however that the data support a narrower (and potentially less interesting) conclusion that individuals with ASD demonstrated an increased influence of recent button presses/motor responses, as the finding which forms the basis of the summary went away when different keys were used to report prior vs. test responses (i.e., in the response invariant condition). We understand that the authors present these data as a challenges to theories of attenuated priors in ASD, but they seem to sidestep the issue that these data make their general conclusion more complicated and weaker. If there was "Increased influence of prior choices on perceptual decisions in autism", as opposed to increased influence of prior motor actions, then the group difference in the response invariant condition should have been replicated.

We agree that the results from the *Response Invariant* condition make interpretation of the primary results more complex, and that they might suggest that the primary results reflect an increased influence of prior motor actions (rather than an increased influence of prior choices). We have responded to this comment in two ways: (1) we now present this limitation in the revised manuscript (please see paragraph 6 in our revised Discussion). (2) We performed further analysis (motivated by the reviewers’ comments below) in order to test whether the results from the primary experiment might reflect a simple motor repetition. Specifically, we hypothesized that a simple motor repetition effect would manifest irrespective of discrimination difficulty – i.e., also for easy discriminations (ones with obvious stimuli). Thus, for a motor repetition effect, we would expect an increase in lapse rates (proportion of incorrect responses for obvious stimuli) in ASD. If, however, lapse rates are not increased in ASD, this might suggest that the increased influence of prior choices in ASD reflects a more complex decision process vs. simple motor repetition. Specifically, that prior choice information is integrated together with current stimulus information. Indeed, lapse rate analysis did not find any significant difference in ASD, in both tasks (please see further details in response to comment #2 below, and new Results section “Comparable lapse rates in ASD” in the revised manuscript). This finding suggests that the prior choice effect in ASD is not just a motor repetition effect. Rather, that information from prior choices is integrated together with current stimulus information as part of a complex decision process, also in ASD.

In light of this point, and the results from the *Response Invariant* condition, we now suggest a more limited and nuanced interpretation of our findings in the revised manuscript – an increased influence of prior choices in ASD when the same actions are used (primary condition), but not when different actions are used (*Response Invariant* condition). In conjunction, these results suggest that the greater influence of prior choices in ASD occurs specifically when perception-action coupling is consistent. This interpretation is in line with the hypothesis that perception and action are intertwined in a perception-action loop (Buckingham et al., 2016) and our recent finding (in a series of related control experiments; Feigin et al., 2021) that sensorimotor (decision-action) coupling boosts the influence of prior choices. Accordingly, the increased influence of prior choices we found in ASD may specifically reflect an increased influence of prior perception-action coupling on subsequent decisions. We have updated the manuscript to reflect this more limited interpretation (please see paragraph 7 in our revised Discussion).

2. Related to the previous point, the word "perseveration", i.e. a tendency to repeat the last key or recent keys is not mentioned. The authors conducted a "response invariant" experiment which showed significant but much smaller biases (Figure 7). Are these significantly smaller than the 1st experiment (as seems to be the case from the plots)? If so, one cannot rule out a major contribution of repeating the recent keys, i.e. perseveration. It would be useful to see the raw data in this case; e.g. what is the %trials of pressing right when the priors were biased to the right. Likely this was high, given that the staircase was symmetric (50/50 trials on left and right) and that a bias emerged from the data.

We agree that motor perseveration could contribute to the results, and accordingly have added this limitation (with specifically the term “motor perseveration”) to our revised Discussion (paragraph 6). However, our finding that ASD have no significant increase in lapse rates (described further below in this comment) suggests that motor perseveration alone does not fully account for our results. Rather, we suggest that prior choice information is weighed together with current sensory information in a complex perceptual decision process that takes into account stimulus strength (unlike simple motor perseveration), also in ASD. Accordingly, prior choices have little effect on easy decisions (also in ASD), and more influence on difficult decisions (where stimulus information is poorer) – and the choice effect is increased in ASD. Taking all the results into account, we suggest that perception-action coupling (which leads to a stronger effect of prior choices in all individuals) is specifically increased in ASD. This is in line with larger choice biases (βprev_choices) in the primary condition vs. the *Response Invariant* condition (*t*_30_ = 2.94, *p* = 0.006, *Cohen’s d* = 0.52, 95% CI = [0.15, 0.89], *t-*test; BF_01_ = 0.15; pooling both groups together), similar to our recent results in adults (Feigin et al., 2021). We have added this comparison to the revised manuscript (please see paragraph 2 in Results section “Effect of recent priors in ASD ensues even when reported using different actions”).

Following the reviewers’ suggestion, we further looked for motor perseveration in the raw data. We considered that calculating the percentages of discriminations for which participants’ responses were in accordance with the prior type (i.e., rightward choices following rightward-biased priors, and leftward choices following left-biased priors) might not be able to dissociate an increased influence of prior choices from a simple motor repetition effect. Both would make the same prediction – an increased percentage of rightward (leftward) choices following rightward (leftward) priors. Therefore, following the rationale suggested here by the reviewers, we further looked for motor perseveration in the raw responses, as follows: we hypothesised that simple motor perseveration should be observed irrespective of stimulus strength (namely, also for obvious/easy stimuli). Accordingly, if our results reflect greater motor perseveration in ASD, one would expect greater lapse rates (a reduced ratio of correct choices for easy discriminations) in ASD. By contrast, an effect of prior choices that is, integrated with current stimulus information might be relatively weak (i.e., no observable effect) for easy (obvious) stimuli. We assessed this in two ways: (i) by comparing lapse rates of ASD vs. controls from the psychometric fits, and (ii) by comparing the percent correct for easy discriminations (i.e., not reliant on the psychometric fits).

Lapse rates did not differ significantly between ASD and controls (raw *p*-values presented below without correction for multiple comparisons), with Bayes factors (slightly) favouring the null hypothesis. Lapse rates (mean ± SEM) for the location discrimination primary condition were 0.030 ± 0.008 and 0.038 ± 0.008 for ASD and controls, respectively (*t*_36_ = -0.74, *p* = 0.47, *Cohen’s d* = 0.24, 95% CI = [-0.88, 0.40], *t-*test; BF_01_ = 2.56). Lapse rates (mean ± SEM) for the location discrimination *Response Invariant* condition were 0.079 ± 0.013 and 0.069 ± 0.015 for ASD and controls, respectively (*t*_30_ = 0.51, *p* = 0.61, *Cohen’s d* = 0.18, 95% CI = [-0.52, 0.87], *t-*test; BF_01_ = 2.69). Lapse rates (mean ± SEM) for the heading discrimination task were 0.015 ± 0.003 and 0.013 ± 0.003 for ASD and controls, respectively (*t*_34_ = 0.5, *p* = 0.62, *Cohen’s d* = 0.08, 95% CI = [-0.006 0.010], *t*-test; BF_01_ = 2.77). Thus no difference in lapse rates was observed in ASD, in both tasks.

To further test this point in a way that does not depend on psychometric fits and the lapse rate parameter, we also calculated and compared the percentage of correct choices for easy discriminations (defined by stimuli with absolute values larger than the 66^th^ percentile, calculated individually for each participant and condition). Also here we found no significant differences between the ASD and control groups (*p*-values without correction for multiple comparisons), with BFs supporting the null hypothesis. Percent correct (mean ± SEM) for the easy discriminations in the location discrimination primary condition was 96.61 ± 0.94% and 95.12 ± 1.16% for ASD and controls, respectively (*t*_36_ = 0.98, *p* = 0.33, *Cohen’s d* = 0.32, 95% CI = [-0.32, 0.96], *t-*test; BF_01_ = 2.17). Percent correct (mean ± SEM) for the easy discriminations in the location discrimination *Response Invariant* condition was 91.21 ± 1.39% and 92.41 ± 1.75%, for ASD and controls, respectively, (*t*_30_ = -0.54, *p* = 0.60, *Cohen’s d* = 0.19, 95% CI = [-0.88, 0.51], *t-*test; BF_01_ = 2.66). Percent correct (mean ± SEM) for the easy discriminations in the heading discrimination task was 88.12 ± 0.74% and 88.86 ± 0.50% for ASD and control group, respectively (*t*_34_ = -0.87, *p* = 0.39, *Cohen’s d* = 0.15, 95% CI = [-2.50 1.00], *t*-test; BF_01_ = 2.28). These extra analyses suggest that our results of an increased influence of prior choices in ASD does not reflect simple motor perseveration. We have added these analysis to the revised manuscript (please see new Results section “Comparable lapse rates in ASD”).

Also the results from another analysis (that we performed in response to comment #4, below) may provide further evidence that the increased influence of prior choices in ASD does not only reflect simple motor perseveration. In that analysis, we investigated the individual contributions of the 5 preceding discriminations to the current decision. Assuming that only the most recent one is relevant for motor preservation (i.e., whether the previous response is repeated), an independent influence of choices preceding the previous discrimination might suggest a compound effect, beyond simple motor perseveration. Indeed, we found significant contributions of choices from before the most recent choice (two choices back, with significantly larger effects in ASD). Please see further details in response to comment #4, below (and new Results section “Prior choice effect seen two steps back” as well as Figure 4—figure supplement 1). We have added this point to the revised Discussion (paragraph 6). In summary, complementary evidence (comparable lapse rates and percent correct for easy discriminations in ASD, and an independent contribution of choices before the immediately preceding choice) suggests that the increased influence of prior choices in ASD goes beyond a simple motor perseveration effect.

3. The finding from the model that the prior stimuli did not have a positive impact (and even negative) on the decision bias is counter-intuitive and needs explanation. There were typically 5 prior trials, ~4 of them on one side, e.g. right, resulting in a higher rate of right presses on the test (because the test was unbiased, and the results showed a bias). Assuming the prior trials were mostly replied correctly, there should be a correlation between the stimuli and the choices. There are two possible reasons why the model produced negative weights – one is that indeed the choices were different from the stimuli, in which case we need to know the performance of the participants on the prior trials (which would be useful anyway). The other possibility is that the choices for the model were binary and the stimuli were continuous. If the stimuli had been coded as binary, it would have been difficult to dissociate between the stimuli and the choices. In this case, the conclusion should be that the prior stimulus laterality could have impacted the test choices, but not their magnitude. This issue should be explained in the text.

A negative influence of prior stimuli on subsequent perception has been documented in many studies (e.g., sensory adaptation /aftereffects, Gibson, 1937; Gibson and Radner, 1937; Anstis et al., 1998; Thompson, P., and Burr, 2009). And, a positive influence of prior choices has also been documented in many studies (Kaneko and Sakai, 2015; Talluri et al., 2018; Feigin et al., 2021). Furthermore, converging evidence from many recent studies suggests that the influence of prior stimuli and prior choices are often in opposite directions – i.e., a negative (adaptive) influence of prior stimuli vs. a positive (consistency bias) influence of prior choices, and that these superimpose (Fritsche et al., 2017; Bosch et al., 2020; Feigin et al., 2021; Sadil et al., 2021). Thus, a slightly negative effect of prior stimuli in the location discrimination experiment is not unique to our data, and is in line with previous results. We have added these points to the revised Discussion (paragraphs 2-3).

Although correlated, prior choices and prior stimuli were separable for the two reasons pointed out by the reviewers: (i) choices are binary, whereas the stimuli are graded, and (ii) participants make mistakes. Regarding the first point, we did not test/model (another) binary parameter of stimulus laterality, because that would imply it has undergone classification (making it ‘choice-like’, and thus difficult to interpret). Rather, we reasoned that a graded stimulus parameter is more reflective of the stimulus (adaptation) effects we aimed to model – in line with traditional sensory adaptation literature that shows affects proportional to stimulus strength (Bosch et al., 2020), rather than to binary stimulus categories. Thus, the basis for the way these parameters were modelled (using graded values for stimulus strength), and the results, are in line with previous studies. We have added these points to the Methods section “Perceptual decision model fit”.

Regarding the second point, participants performed well, but still made mistakes. Percent correct (mean ± SEM) for the ‘prior’ discriminations in the location discrimination primary condition was 88.33 ± 1.09% and 87.56 ± 1.54 % for ASD and controls, respectively (with no significant difference between the groups, *t*_36_ = 0.40, *p* = 0.69, *Cohen’s d* = 0.13, 95% CI = [-0.51, 0.77], *t-*test; BF_01_ = 2.98). Percent correct (mean ± SEM) for the ‘prior’ discriminations in the *Response Invariant* condition was 81.26 ± 1.20% and 84.57 ± 2.08% for ASD and controls, respectively (with no significant difference between the groups, *t*_30_ = -0.52, *p* = 0.6, *Cohen’s d* = 0.18, 95% CI = [-0.88, 0.51], *t-*test; BF_01_ = 2.68). Thus, separability in modelling of the prior choice and prior stimuli effects results from a combination of these two differences (i.e., choices are binary while stimuli are continuous, and participants make mistakes). Overall, Pearson correlations between stimuli and choices (across all stimuli in the block) ranged on average 0.61 – 0.70 (depending on condition), i.e., R^2^ ≲ 0.5 (this point was added to the Methods section “Perceptual decision model fit”). Accordingly, stimulus and choice parameters contained unique information, and we used the modelling approach to identify these different effects, in accordance with previous studies.

4. Currently, all previous trials are treated equal in the model, but their contribution is not necessarily equal. Could you analyze the data to assess whether there is a differential contribution of preceding trials, i.e. the details of the serial dependency.

In response to this comment we reran the regression analysis, modified to study the separate contributions of the five preceding ‘priors’. There were two potential obstacles to performing this analysis: a) the number of priors for a given trial was random (µ = 5) so there could be fewer than five (there could even be only one prior). b) Modelling both prior choices and prior stimuli five steps back means 10 ‘prior’ parameters. This could potentially lead to overfitting and spurious results (especially for the more distant ones, where the prior effect is presumably weaker). To overcome these issues we took the following two measures: (i) we fit the model to all the data (prior and test, viewed as one long sequence of trials) using the five preceding steps as ‘priors’ (irrespective of whether they were initially termed ‘prior’ or ‘test’). This was possible in the primary location discrimination condition because the stimulus color and task (i.e., which button to use) were the same for both prior and test circles (unlike the *Response Invariant* condition). (ii) We first looked whether the effect of prior choices and prior stimuli were significant for the most recent two steps (*t*-1 and *t*-2): both prior choices were significant (*p* < 6·10^-5^; *t*-tests) and both prior stimuli were not (*p* > 0.14; *t*-tests). Thus, to prevent overfitting/ spurious results, we investigated further only the prior choice effect for the five preceding steps (please see Figure 4—figure supplement 1).

We found that two choices back (*t*-1 and *t*-2) significantly (positively) affected decisions (for one step back (*t*-1): *t*_17_ = 7.16, *p* = 1.6·10^-6^*, Cohen’s d* = 1.69, 95% CI = [0.95, 2.41] for the ASD group and *t*_19_ = 4.46, *p* = 0.0003, *Cohen’s d* = 1.00, 95% CI = [0.45, 1.53] for the control group. For two steps back (*t*-2): *t*_17_ = 9.09, *p* = 6·10^-8^, *Cohen’s d* = 2.14, 95% CI = [1.28, 2.98] for the ASD group and *t*_19_ = 4.63, *p* = 0.0002, *Cohen’s d* = 1.03, 95% CI = [0.48, 1.57] for the control group, *t*-tests). Beyond that (*t*-3, *t*-4, *t*-5) the effects were not significant (*p* > 0.34; *t*-tests). The choices one step back (*t*-1) had a stronger effect vs. those two steps back (*t*-2) (*F*_1,36_ = 8.04, *p* = 0.007, *η_p_²* = 0.18; mixed ANOVA). The overall effect of prior choices was stronger in the ASD group compared to the control group (*F*_1,36_ = 9.46, *p* = 0.004, *η_p_²* = 0.21; mixed ANOVA). Pairwise comparisons between ASD and controls for the choice effect from one (*t*-1) and two (*t*-2) steps back (separately) found a significant difference at *t*-2 (*F*_1,36_ = 12.62, *p* = 0.001, *η_p_²* = 0.26; mixed ANOVA), with a similar trend (but not significant) at *t*-1 (*F*_1,36_ = 2.65, *p* = 0.11, *η_p_²* = 0.07; mixed ANOVA). Please see Figure 4—figure supplement 1 and new Results section “Prior choice effect seen two steps back” in the revised manuscript.

5. The performance on the test trials staircase procedure is not reported, only the PSE difference. Thresholds and slopes, indicators of sensitivity and precision of performance in the task are not detailed. Bias magnitude could vary as a factor of noise or sensitivity.

In response to this comment we have added additional comparisons of performance between the ASD and control groups. Specifically (in addition to comparing βcurr_stimulus, which is a measure of sensitivity) we now also compare thresholds, which are inversely proportional to sensitivity. Please see our specific responses in the subsections of this question (a-f) below.

a. Biases are likely to push the staircase procedure to higher laterality discrimination thresholds. Could worse performance (more errors) on the staircase procedure amplify (but not create) the bias?

The staircase procedure only sets the distribution of the stimuli presented. As long as this distribution covers an adequate range of stimuli (i.e., it has enough difficult trials, and enough easy trials) and is balanced on both sides (right and left), biases and thresholds measured should theoretically be unaffected by this distribution. For example, the “method of constant stimuli”, which provides the same number trials for each possible stimulus value (and therefore more easy trials vs. a staircase), is a common and valid (albeit less efficient) way to measure biases and thresholds. The staircases here achieved their target range of ~70.7% correct. Thus, we do not believe that a slight increase in the number of easy trials that can result from a bias (which was small relative to the range of stimuli) with counterbalanced trials and prior biasing would affect the biases or thresholds measured. We have added this point to the revised Methods section “Location discrimination task – primary condition” (paragraphs 4-6).

b. Was sensitivity considered in anyway in the analysis?

The βcurr_stimulus parameter (presented in the paper) is a measure of sensitivity. Specifically, larger values mean higher stimulus sensitivity. For this measure we found no significant differences between the groups in the location discrimination primary condition, *Response Invariant* condition and heading discrimination. Please see their respective Results sections: “Enhanced influence of recent choices in ASD for location discrimination” (last paragraph), “Enhanced influence of recent choices in ASD for heading discrimination” (last paragraph), and “Effect of recent priors in ASD ensues even when reported using different actions” (paragraph 3). In order to answer this comment also in terms of the psychometric functions we have now added additional comparisons. Specifically, we now compare thresholds (i.e., σ extracted from the psychometric functions). Please see the respective Results sections (referenced above) to which we have added these results, and our further responses to the comments below.

c. Did performance differ between groups? Example psychometric curves shown seem shallower in ASD.

We tested for differences in performance sensitivity by comparing the βcurr_stimulus parameters between the groups (finding no significant differences, please see the last paragraph in Results section “Enhanced influence of recent choices in ASD for location discrimination”). Furthermore, we also tested for differences in performance between the ASD and control groups by comparing the thresholds extracted from the psychometric fits. To reduce unwanted effects of ‘priors’ on the threshold estimate, psychometric curves were fit separately, per prior type, and the thresholds averaged (per participant). Because thresholds are non-negative and scale logarithmically, averaging and statistical comparisons were performed on the natural log values (we have added these details to the updated Methods section “Data analysis – psychometric fits”). We think that the model parameter (βcurr_stimulus) may offer a better measure of sensitivity because it reflects a fit to all the data (vs. dividing the data and therefore estimating thresholds from fewer trials). But, we now also present the threshold results.

No significant differences in threshold were found between the groups (uncorrected *p*-values presented below) with Bayes factors (slightly) favouring the null hypothesis. We have added these comparisons to the respective Results sections in the manuscript. In the location discrimination task primary condition, the thresholds (natural log values, mean ± SEM) were 0.78 ± 0.13 and 0.58 ± 0.17 for ASD and controls, respectively. These did not differ significantly between groups (*t*_36_ = 0.96, *p* = 0.34, *Cohen’s d* = 0.31, 95% CI = [-0.33,0.95], *t*-test; BF_01_ = 2.21). For the *Response Invariant* condition, these values were 0.68 ± 0.07 and 0.52 ± 0.13 (for ASD and controls, respectively). These too did not differ significantly between groups (*t*_30_ = 1.02, *p* = 0.32, *Cohen’s d* = 0.36, 95% CI = [-0.34, 1.06], *t*-test; BF_01_ = 2.00).

In our previous comparison of heading discrimination thresholds (Zaidel et al., 2015) ASD and controls had comparable vestibular thresholds, and visual thresholds depended on the noise condition (ASD had slightly better and worse performance for 100% visual coherence, and noisy conditions, respectively) – on average, however, they were comparable. But, the threshold comparisons in that study did not take into account the increased influence of prior choices in ASD (and used an older version of the psignifit software). Hence, we compared them again here, sorting the data by prior choice (separate psychometric fits) and then averaging the thresholds (per participant and condition) to attain a better estimate of performance. For the heading discrimination task, the thresholds (natural log values, mean ± SEM) over all conditions and coherences (pooled) were 0.81 ± 0.09 for the ASD group and 0.71 ± 0.05 for the control group, and they did not differ significantly (overall) between groups (*p* = 0.78, *η_p_^2^* = 0.003 mixed ANOVA; BF_01_ = 3.25). Also, there was no significant difference in the model estimate of performance (βcurr_stimulus) between ASD and controls (Figure 6 in the manuscript). Hence, also in the heading discrimination data, the increased influence of prior choices ASD does not likely result from differences in performance.

d. Was performance correlated with bias?

Psychometric curve ΔPSEs might be expected to correlate with psychometric thresholds, for technical reasons. That is because low sensitivity (i.e., large thresholds) would allow the previous choice effect to be *relatively* larger vs. the reduced effect of the current stimulus (for low sensitivity). However, when measuring bias from the logistic regression model, this is not the case. Here, the influence of prior choices is quantified by a dedicated latent parameter (βprev_choices), which is independent of the stimulus effect (βcurr_stimulus). This allows fitting the data with the participant’s individual perceptual sensitivity, and independently estimating their prior choice effect (according to Equation 5 from the manuscript). This is one of the advantages and reasons for the using the logistic regression model. For example, two participants with the same prior choice effect, but different stimulus sensitivities (which could result in a larger ΔPSE for the participant with lower sensitivity) will be correctly fit by the model: i.e., lower vs. higher βcurr_stimulus values, but similar values for βprev_choices. Therefore, there is no theoretical expectation that βprev_choices will correlate with performance for technical/trivial reasons. Beyond this theoretical explanation, we also empirically tested for correlations between prior choice effects and sensitivity for both measures: (i) ΔPSEs vs. thresholds, and (ii) model estimates βprev_choices vs. βcurr_stimulus (pooling ASD and control groups, uncorrected *p*-values presented below).

In line with the theoretical expectation (described above) ΔPSEs do seem to correlate with thresholds. This was significant for the location discrimination primary condition (*r_36_* = 0.48, *p* = 0.002), although a significant correlation was not seen for the *Response Invariant* condition (*r_30_* = -0.27, *p* = 0.13). By contrast the model estimated prior choice effect (βprev_choices) did not significantly correlate with sensitivity (βcurr_stimulus) for the location discrimination primary condition (*r_36_* = -0.18, *p* = 0.27), and for the *Response Invariant* condition (*r_30_* = -0.27, *p* = 0.13). These correlation results (between the model parameters βcurr_stimulus and βprev_choices) have been added to the Results sections: “Enhanced influence of recent choices in ASD for location discrimination” (last paragraph), and “Effect of recent priors in ASD ensues even when reported using different actions” (paragraph 3) for the primary and *Response Invariant* conditions, respectively.

In the heading discrimination task, there was no significant correlation between thresholds and ΔPSE: *r*_34_ = -0.07, *p* = 0.67. Similarly, no significant correlation was found between βprev_choices and βcurr_stimulus (*r*_34_ = 0.06, *p* = 0.74). This correlation result (between the model parameters βcurr_stimulus and βprev_choices) has been added to the Results section: “Enhanced influence of recent choices in ASD for heading discrimination” (last paragraph).

In conclusion, although the ΔPSE measure can correlate with sensitivity, sensitivity did not significantly differ between ASDs and controls (see response to comment #5c). Therefore, differences in sensitivity do not likely account for the different prior choice effects observed in ASD (even for the ΔPSE measure). Furthermore, our conclusions in this manuscript rely on the model extracted prior choice effect, which does not correlate (theoretically and empirically) with sensitivity.

e. Could larger consistency bias in the ASD group result from weaker performance, more lapses of attention?

We did not find significant differences in performance between the ASD and the control groups, in both tasks (when comparing thresholds, and βcurr_stimulus, please see response to comment #5c above). We also did not find any significant differences in lapse rates between ASD and controls (please see response to comment #2 above). Finally, we did not find significant correlations between βprev_choices and βcurr_stimulus (please see response to comment #5d above). Thus, the larger influence of prior choices in ASD group does not likely result from weaker performance or lapses.

f. How would the model fitting look like and how would it interact with the biases?

The advantage of using the model to measure the prior choice influence is that the model parameters (βprev_choices and βcurr_stimulus) measure their individual contributions to the choices. Hence we believe that the findings of increased influence of prior choices in ASD, are specific and do not reflect other differences in task performance.

References:

Anstis S, Verstraten FAJ, Mather G (1998) The motion aftereffect. Trends Cogn Sci 2:111–117.

Banks MS, Gepshtein S, Landy MS (2004) Why Is Spatial Stereoresolution so Low? J Neurosci 24:2077–2089.

Bosch E, Fritsche M, Ehinger B V., de Lange FP (2020) Opposite effects of choice history and evidence history resolve a paradox of sequential choice bias. J Vis 20:1–13.

Buckingham G, Michelakakis EE, Rajendran G (2016) The Influence of Prior Knowledge on Perception and Action: Relationships to Autistic Traits. J Autism Dev Disord 46:1716–1724.

Burge J, Fowlkes CC, Banks MS (2010) Natural-scene statistics predict how the figure-ground cue of convexity affects human depth perception. J Neurosci 30:7269–7280.

Burton N, Jeffery L, Bonner J, Rhodes G (2016) The timecourse of expression aftereffects. J Vis 16:1–1.

Dannemiller JL, Banks MS (1983) The development of light adaptation in human infants. Vision Res 23:599–609.

Feigin H, Baror S, Bar M, Zaidel A (2021) Perceptual decisions are biased toward relevant prior choices. Sci Rep 11:648.

Felsen G, Shen YS, Yao H, Spor G, Li C, Dan Y (2002) Dynamic modification of cortical orientation tuning mediated by recurrent connections. Neuron 36:945–954.

Fritsche M, Mostert P, de Lange FP (2017) Opposite Effects of Recent History on Perception and Decision. Curr Biol 27:590–595.

García-Pérez FJ, Lario Y, Fernández-López J, Sayas E, Pérez-Alvarez JA, Sendra E (2005) Effect of orange fiber addition on yogurt color during fermentation and cold storage. Color Res Appl 30:457–463.

Gibson JJ (1937) Adaptation with negative after-effect. Psychol Rev 44:222–244.

Gibson JJ, Radner M (1937) Adaptation, after-effect and contrast in the perception of tilted lines. J Exp Psychol 20:453–467.

Johnson M, Suengas A, Foley MA, Raye C (1988) Phenomenal Characteristics of Memories for Perceived and Imagined Autobiographical Events. J Exp Psychol Gen 117:371–376.

Kaneko Y, Sakai K (2015) Dissociation in decision bias mechanism between probabilistic information and previous decision. Front Hum Neurosci 9:1–11.

Landy MS, Kojima H (2001) Ideal cue combination for localizing texture-defined edges. J Opt Soc Am A 18:2307.

Leek MR (2001) Adaptive procedures in psychophysical research. Percept Psychophys 63:1279–1292.

MacNeilage PR, Banks MS, DeAngelis GC, Angelaki DE (2010) Vestibular heading discrimination and sensitivity to linear acceleration in head and world coordinates. J Neurosci 30:9084–9094 Available at: http://www.ncbi.nlm.nih.gov/pubmed/20610742 [Accessed October 27, 2019].

Nilsson ME, Schenkman BN (2016) Blind people are more sensitive than sighted people to binaural sound-location cues, particularly inter-aural level differences. Hear Res 332:223–232.

Pegors TK, Mattar MG, Bryan PB, Epstein RA (2015) Simultaneous perceptual and response biases on sequential face attractiveness judgments. J Exp Psychol Gen 144:664–673 Available at: https://pubmed.ncbi.nlm.nih.gov/25867223/ [Accessed July 26, 2020].

Rhodes G, Jeffery L, Clifford CWG, Leopold DA (2007) The timecourse of higher-level face aftereffects. Vision Res 47:2291–2296.

Sadil P, Cowell R, Huber DE (2021) The Yin-yang of Serial Dependence Effects: Every Response is both an Attraction to the Prior Response and a Repulsion from the Prior Stimulus. Available at: https://psyarxiv.com/f52yz/.

Samson F, Mottron L, Soulières I, Zeffiro TA (2012) Enhanced visual functioning in autism: An ALE meta-analysis. Hum Brain Mapp 33:1553–1581.

Sekuler R, Littlejohn J (1974) Tilt aftereffect following very brief exposures. Vision Res 14:151–152.

Sinha P, Kjelgaard MM, Gandhi TK, Tsourides K, Cardinaux AL, Pantazis D, Diamond SP, Held RM (2014) Autism as a disorder of prediction. Proc Natl Acad Sci U S A 111:15220–15225.

Sou KL, Xu H (2019) Brief facial emotion aftereffect occurs earlier for angry than happy adaptation. Vision Res 162:35–42.

Stewart CR, Sanchez SS, Grenesko EL, Brown CM, Chen CP, Keehn B, Velasquez F, Lincoln AJ, Müller RA (2016) Sensory Symptoms and Processing of Nonverbal Auditory and Visual Stimuli in Children with Autism Spectrum Disorder. J Autism Dev Disord 46:1590–1601.

Suzuki S (2001) Attention-dependent brief adaptation to contour orientation: A high-level aftereffect for convexity? Vision Res 41:3883–3902.

Talluri BC, Urai AE, Tsetsos K, Usher M, Donner TH (2018) Confirmation Bias through Selective Overweighting of Choice-Consistent Evidence. Curr Biol 28:3128–3135.

Thompson, P., and Burr D (2009) Visual aftereffects. Curr Biol 19:R11–R14.

Xu H, Liu P (2012) Adapting to an incomplete curve generates the same curvature aftereffect as a complete curve. J Vis 12:1052–1052.

Zaidel A, Goin-Kochel RP, Angelaki DE (2015) Self-motion perception in autism is compromised by visual noise but integrated optimally across multiple senses. Proc Natl Acad Sci U S A 112:6461–6466 Available at: http://www.ncbi.nlm.nih.gov/pubmed/25941373.